# FliH and FliI help FlhA bring strict order to flagellar protein export in *Salmonella*
Miki Kinoshita [1,4], Tohru Minamino [1,4] ✉, Takayuki Uchihashi [2] & Keiichi Namba [1,3]

The flagellar type III secretion system (fT3SS) switches substrate specificity from rod-hook-type to filament-type upon hook completion, terminating hook assembly and initiating filament assembly. The C-terminal cytoplasmic domain of FlhA (FlhA$_C$) forms a homo-nonameric ring and is directly involved in substrate recognition, allowing the fT3SS to coordinate flagellar protein export with assembly. The highly conserved GYXLI motif (residues 368–372) of FlhA$_C$ induces dynamic domain motions of FlhA$_C$ required for efficient and robust flagellar protein export by the fT3SS, but it remains unknown whether this motif is also important for ordered protein export by the fT3SS. Here we analyzed two GYXLI mutants, *flhA(GAAAA)* and *flhA(GGGGG)*, and provide evidence suggesting that the GYXLI motif in FlhA$_C$ requires the flagellar ATPase complex not only to efficiently remodel the FlhA$_C$ ring structure for the substrate specificity switching but also to correct substrate recognition errors that occur during flagellar assembly.

The flagellum of *Salmonella enterica* serovar Typhimurium (hereafter referred to as *Salmonella*) is a motility organelle consisting of the basal body acting as a rotary motor, the filament as a helical propeller, and the hook as a universal joint connecting them to transmit motor torque to the filament. Flagellar assembly begins with the basal body, followed by the hook and finally the filament (Fig. 1)[1,2]. To construct flagella, the flagellar type III secretion system (fT3SS), located at the base of each flagellum, transports flagellar structural subunits from the cytoplasm to the distal end of the growing flagellar structure. The fT3SS consists of a transmembrane export gate complex made of FlhA, FlhB, FliP, FliQ, and FliR and a cytoplasmic ATPase complex consisting of FliH, FliI, and FliJ (Fig. 1)[3,4]. The export gate complex uses both H$^+$ and Na$^+$ as the coupling ion and acts as a cation/protein antiporter that couples inward-directed cation flow with outward-directed protein translocation[5–11]. The cytoplasmic ATPase complex acts not only as an ATP-driven activator of the export gate complex but also as a dynamic carrier that delivers export substrates and chaperone-substrate complexes from the cytoplasm to the export gate complex[12,13].

Flagellar structural subunits fall into two distinct, rod-hook-type (hereafter referred to as RH-type) and filament-type (hereafter referred to as F-type) classes based on the export substrate specificity of the fT3SS[14]. During hook-basal body (HBB) assembly, the fT3SS specifically acts on the RH-type substrates needed for assembly of the rod and hook. Four F-type substrates, FlgK, FlgL, FlgM, and FliD, are expressed during HBB assembly, but are not transported[15]. Once the HBB is complete, the fT3SS switches

substrate specificity from the RH-type to the F-type to build the filament at the hook tip (Fig. 1). The fT3SS uses a secreted molecular ruler named FliK not only to measure the hook length but also to catalyze the substrate specificity switching of the fT3SS when the hook length reaches about 55 nm[16–20]. The export switch of the fT3SS consists of the C-terminal cytoplasmic domains of FlhA (FlhA$_C$) and FlhB (FlhB$_C$)[21,22]. A direct interaction between FliK and FlhB$_C$ causes a conformational change in FlhB$_C$, followed by conformational rearrangements of FlhA$_C$ that terminates hook assembly and initiates filament assembly[23–25]. Therefore, in *Salmonella* strains with loss-of-function of FliK or specific amino acid substitutions in FlhA$_C$ or FlhB$_C$, the substrate recognition mode of the fT3SS remains in the RH state, resulting in unusually elongated hooks named polyhooks.

FlhA$_C$ consists of four domains, D1, D2, D3, and D4, and a flexible linker (FlhA$_L$) connecting FlhA$_C$ with the N-terminal transmembrane domain of FlhA (Fig. 2a)[26]. FlhA$_C$ forms a nonameric ring in the fT3SS (Fig. 1)[27]. The highly conserved Asp-456 and Thr-490 residues are located within a hydrophobic dimple formed by the relatively well-conserved Leu-438, Ile-440, Pro-442, Phe-459, Leu-461, Val-482, and Val-487 residues (Fig. 2a). This conserved dimple, located at the interface between domains D1 and D2, is involved in substrate recognition[28–30]. The C-terminal region of FlhA$_L$ (FlhA$_{L-C}$) including Glu-351, Trp-354, and Asp-356 acts as a switch to induce the structural transition of the FlhA$_C$ ring from the RH state to the F state (Fig. 1). During HBB assembly, FlhA$_{L-C}$ binds to the conserved

[1]Graduate School of Frontier Biosciences, Osaka University, 1-3 Yamadaoka, Suita, Osaka 565-0871, Japan. [2]Department of Physics, Nagoya University, Chikusa-ku, Nagoya 464-8602, Japan. [3]JEOL YOKOGUSHI Research Alliance Laboratories, Osaka University, 1-3 Yamadaoka, Suita, Osaka 565-0871, Japan. [4]These authors contributed equally: Miki Kinoshita, Tohru Minamino. ✉e-mail: tohru@fbs.osaka-u.ac.jp

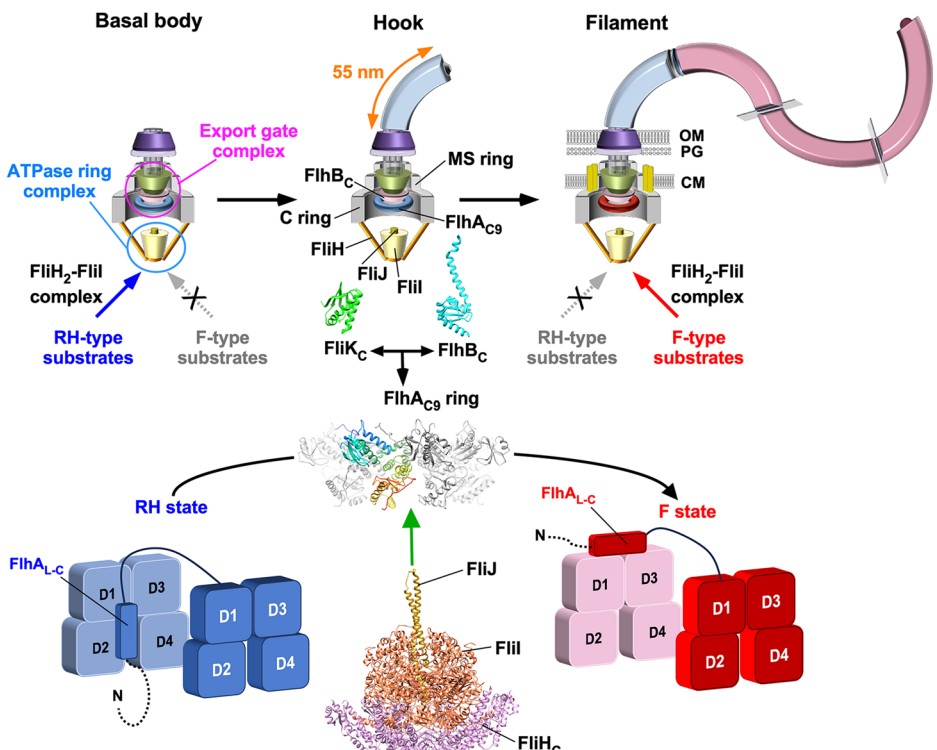

**Fig. 1 | Model for substrate specificity switching of the flagellar type III secretion system.** Flagellar assembly begins with the basal body, followed by the hook and finally the filament. The flagellar type III secretion system (fT3SS) consists of a transmembrane export gate complex and a cytoplasmic ATPase ring complex consisting of FliH, FliI, and FliJ. FliH and FliI also exist as a cytoplasmic $FliH_2FliI$ complex acting a dynamic carrier to deliver export substrates and chaperone-substrate complexes to the export gate complex. The export gate complex is located within the MS ring. The C-terminal cytoplasmic domains of FlhA ($FlhA_C$) forms a nonameric ring ($FlhA_{C9}$) that projects into the central cavity of the C ring. The ATPase ring complex associates with the C ring. The fT3SS transports 14 different flagellar proteins during flagellar assembly, and these 14 proteins are classified into two distinct, RH-type and F-type export classes based on the substrate specificity of the fT3SS. The $FlhA_C$ ring takes at least two distinct conformational states: one is the RH state, in which the C-terminal region of the flexible linker of FlhA ($FlhA_{L-C}$)

binds to a well-conserved hydrophobic dimple located at an interface between domains D1 and D2 (blue); and the other is the F state, in which $FlhA_{L-C}$ binds to the D1 and D3 domains of the nearest $FlhA_C$ subunit (red). During hook-basal body assembly, the fT3SS transports RH-type substrates but not F-type substrates. When the hook length reaches about 55 nm, an interaction between the C-terminal domain of FliK ($FliK_C$) and $FlhB_C$ induces a detachment of $FlhA_{L-C}$ from the hydrophobic dimple and its attachment to the D1 and D2 domains, allowing the fT3SS to terminate RH-type protein export and initiate F-type protein export. The present study establishes that the cytoplasmic ATPase complex is required for the $FlhA_C$ ring to efficiently switch its conformation from the RH state to the F state. Atomic models of $FliK_C$ (PDB ID: 2RRL), $FlhB_C$ (PDB ID: 3B0Z), $FlhA_C$ (PDB ID: 3A5I), the $FliH_{C2}$-FliI complex (PDB ID: 5B0O), and FliJ (PDB ID: 3AJW) are shown in Cα ribbon representation.

hydrophobic dimple, not only facilitating the export of the hook protein (FlgE) but also suppressing the interaction of $FlhA_C$ with flagellar export chaperones (FlgN, FliS, FliT) in complex with their cognate F-type substrates[31]. Upon hook completion, $FlhA_{L-C}$ dissociates from the conserved dimple and binds to the D1 and D3 domains of its neighboring $FlhA_C$ subunit in the ring, terminating RH-type protein export and initiating F-type protein export[22,31,32] (Fig. 1). However, the structural remodeling mechanism of the $FlhA_C$ ring remains a mystery.

Gly-368 of *Salmonella* FlhA is located within the highly conserved GYXLI motif in domain D1 of $FlhA_C$ and is important for dynamic domain motions required for flagellar protein export (Fig. 2a)[33,34]. The temperature-sensitive *flhA(G368C)* mutant cultured at 42 °C produces no flagella but at 30 °C produces almost the same number of flagella as wild-type cells[35], and its average hook length is comparable to that of the wild-type[33]. In contrast, the *ΔfliH-fliI flhB(P28T)* strain, which produces a few flagella even in the absence of FliH and FliI[5], produces no filaments even at 30 °C when the *flhA(G368C)* mutation is added[33]. Furthermore, the average hook length of the *ΔfliH-fliI flhB(P28T) flhA(G368C)* strain is longer than that of the *ΔfliH-fliI flhB(P28T)* strain despite the presence of FliK[33]. Because the *flhB(P28T)* mutation alone does not affect the hook length control at all[36], FliH and FliI would be required for a conformational change of the conserved GYXLI motif to induce the structural transition of the $FlhA_C$ ring from the RH state to the F state.

Here, to clarify this possibility, we analyzed the Y369A/R370A/L371A/I372A (hereafter referred to as AAAA) and Y369G/R370G/L371G/I372G (hereafter referred to as GGGG) mutants and provide evidence that the conserved GYXLI motif is important for ordered protein export by the fT3SS.

## Results
### Effect of the AAAA and GGGG mutations on flagellar protein export
The AAAA and GGGG mutations have been shown to inhibit flagella-driven motility[34]. To address how these mutations affect flagellar formation, we analyzed the secretion levels of RH-type proteins, such as FlgD, FlgE, and FliK, and F-type proteins, such as FlgM, FlgK, FlgL, and FliC. The levels of FlgD, FlgE, and FliK secreted by the AAAA and GGGG mutants were higher than the wild-type levels (Fig. 2b). Consistently, these two mutants produced polyhooks, and the average polyhook lengths of the AAAA and GGGG mutants were 225.7 ± 176.8 (mean ± SD) nm (N = 300) and 471.1 ± 285.1 nm (N = 300), respectively (Fig. 2c). In contrast to the RH-type substrates, the AAAA mutation reduced the secretion levels of FlgM, FlgK, FlgL, and FliC whereas the GGGG mutation inhibited the secretion of these four F-type proteins (Fig. 2b). Consistently, the AAAA mutant produced short filaments whereas the GGGG mutant produced no filaments (Supplementary Fig. 1). Because these two mutants produce polyhooks even

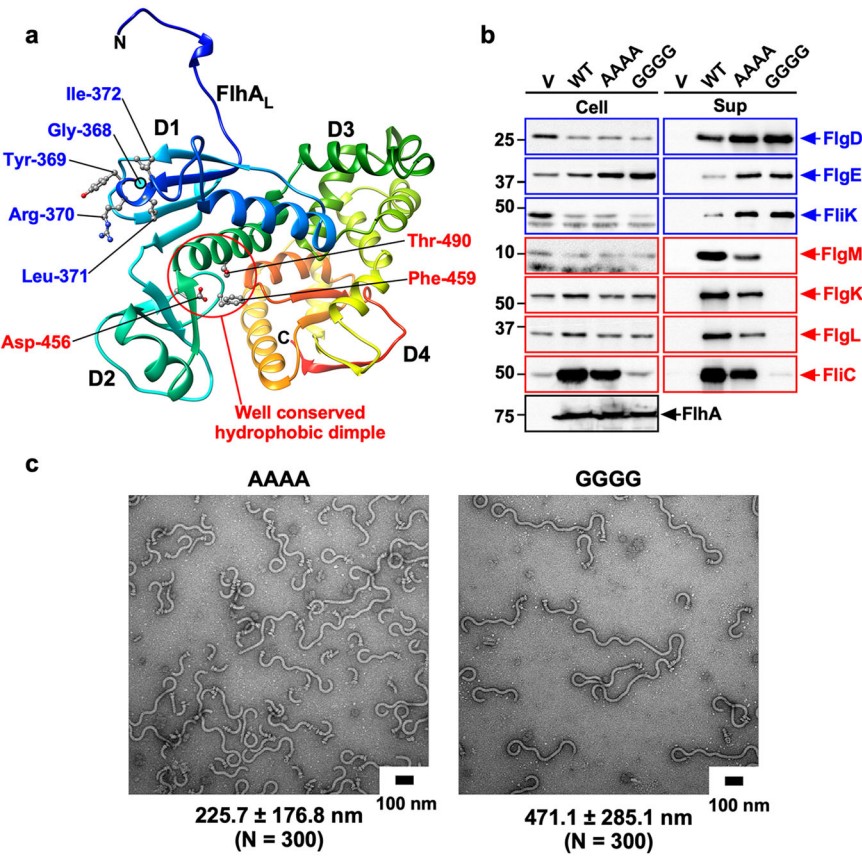

**Fig. 2 | Effect of the AAAA and GGGG mutations on flagellar protein export and assembly. a** Structural model of the FlhA$_C$ monomer (PDB ID: 3A5I). FlhA$_C$ consists of four domains, D1, D2, D3 and D4, and a flexible linker (FlhA$_L$). A highly conserved Gly-368 residue (cyan circle) in domain D1 forms the conserved GYXLI motif along with Tyr-369, Arg-370, Leu-371, and Ile-372, which forms a short α-helix. The well-conserved dimple including the Asp-456, Phe-459, and Thr-490 residues is responsible for the interaction of FlhA$_C$ with flagellar export chaperones in complex with F-type substrates. The Cα backbone is color-coded from blue to red, going through the rainbow colors from the N-terminus to the C-terminus. **b** Secretion analysis of FlgD, FlgE, FliK, FlgM, FlgK, FlgL, and FliC by immunoblotting. Whole cell proteins (Cell) and culture supernatants (Sup) were prepared from the *Salmonella* NH001 (Δ*flhA*) strain transformed with pTrc99AFF4 (indicated as V), pMM130 (indicated as WT), pMKM130-A4 (indicated as AAAA), or pMKM130-G4 (indicated as GGGG). A 3 µl solution of each sample normalized to an optical density of OD$_{600}$ was subjected to SDS-PAGE and analyzed by immunoblotting using polyclonal anti-FlgD (1st row), anti-FlgE (2nd row), anti-FliK (3rd row), anti-FlgM (4th row), anti-FlgK (5th row), anti-FlgL (6th row), anti-FliC (7th row), or anti-FlhA$_C$ (8th row) antibody. RH-type and F-type substrates are highlighted in blue and red, respectively. Molecular mass markers (kDa) are shown on the left. The regions of interest were cropped from original immunoblots shown in Supplementary Fig. 8. **c** Electron micrographs of polyhook-basal bodies isolated from the AAAA and GGGG mutants. The average polyhook length and standard deviations are shown. N indices the number of polyhook-basal bodies that were measured.

when both FliK and FlhB are intact, we suggest that the substrate specificity of fT3SS is determined by the conformational state of the FlhA$_C$ ring.

### Isolation of up-motile mutants from the AAAA and GGGG mutants

To clarify how the AAAA and GGGG mutations severely impair or inhibit the conformational transition of the FlhA$_C$ ring from the RH state to the F state, we isolated seven and three up-motile mutants from the AAAA and GGGG mutants, respectively (Fig. 3a, left panel and Fig. 4a). DNA sequencing of the seven up-motile mutants isolated from the AAAA mutant identified two missense mutations, A372V (isolated three times) and A372T in FlhA$_C$ (Fig. 3b), and two missense mutations, Q338R and A405V (isolated twice) in FliK (Fig. 4b). The three up-motile mutants isolated from the GGGG mutant revealed that all suppressor mutations were G372V missense mutations in FlhA$_C$ (Fig. 3b). All the intragenic suppressor mutations share a common feature that the change of residue occurred at position 4 of the four mutated residues in the parent *flhA* mutants, such as from AAAA to AAAV or AAAT and from GGGG to GGGV. This suggests that the hydrophobic side chain of Ile-372 is the most important for the export switching function of FlhA$_C$.

To confirm this, we constructed the *flhA(I372A)* and *flhA(I372G)* mutants. The *flhA(I372A)* mutation reduced the motility in soft agar

whereas the *flhA(I372G)* mutation inhibited the motility (Fig. 5a, left panel). The levels of FlgD and FlgE secreted by the *flhA(I372A)* mutant was slightly higher than the wild-type levels, (Fig. 5b, left panels). Consistently, the *flhA(I327A)* mutant produced longer hooks in addition to normal hooks (Fig. 5c). The *flhA(I372A)* mutation did not affect the secretion of FlgK, FlgL, and FliC at all. On the other hand, the *flhA(I372G)* mutation increased the secretion levels of FlgD and FliK and reduced the secretion levels of FlgK, FlgL, and FliC compared to wild-type cells (Fig. 5b, left panels). Consistently, the *flhA(I372G)* mutant produced polyhooks (Fig. 5c). Because the GYXLI motif forms a short α-helix that adopts distinct conformations in the FlhA$_C$ crystal structures (Supplementary Fig. 2), we suggest that a proper conformational change of the GYXLI motif is critical for efficient and robust structural transition of the FlhA$_C$ ring from the RH state to the F state.

### Characterization of intragenic AAAV, AAAT, and GGGV suppressor mutants

To examine whether intragenic suppressor mutations shorten the length of polyhooks produced by the AAAA and GGGG mutants, we purified HBBs from the AAAV, AAAT, and GGGV mutants and measured their hook length (Fig. 3c, upper panel). The hook lengths of the AAAV, AAAT, and GGGV mutants were 46.8 ± 6.4 nm (*N* = 300), 56.1 ± 14.7 nm (*N* = 300),

**Fig. 3 | Effect of mutations in the conserved GYXLI motif of FlhA on flagellar protein export and assembly in the presence and absence of FliH and FliI. a** Motility of the *Salmonella* NH001 (*ΔflhA*, indicated as ΔA) or NH003 [*ΔfliH-fliI flhB(P28T) ΔflhA*, indicated as ΔHI-B* ΔA] strain transformed with pTrc99AFF4 (indicated as V), pMM130 (indicated as WT), pMKM130-A4 (indicated as AAAA), pMKM130-A3V (indicated as AAAV), pMKM130-A3T (indicated as AAAT), pMKM130-G4 (indicated as GGGG), or pMKM130-G3V (indicated as GGGV) in the presence (left panel) and absence (right panel) of FliH and FliI. Soft tryptone agar plates were incubated at 30 °C for 7 hours (left panel) or 18 hours (right panel). **b** Location of intragenic suppressor mutations isolated from the AAAA and GGGG mutants. The conserved GYXLI motif of FlhA is highlighted in cyan. The intragenic A372V or A372T suppressor mutation isolated from the AAAA mutant is the change of alanine at position 4 in the AAAA sequence to valine or threonine, respectively, and the intragenic G372V suppressor mutation is the change of glycine at position 4 in the GGGG sequence to valine. **c** Electron micrographs of hook-basal bodies isolated from the above transformants. The average hook length and standard deviations are shown. N indices the number of hook-basal bodies and polyhook-basal bodies that were measured. **d** Secretion assays of flagellar proteins. Immunoblot, using polyclonal anti-FlgD (1st row), anti-FlgE (2nd row), anti-FliK (3rd row), anti-FlgK (4th row), anti-FlgL (5th row), anti-FliC (6th row), anti-FliD (7th row), or anti-FlhAC (8th row) antibody, of whole cell proteins (Cell) and culture supernatants (Sup) prepared from the above transformants. RH-type and F-type substrates are highlighted in blue and red, respectively. Molecular mass markers (kDa) are shown on the left. The regions of interest were cropped from original immunoblots shown in Supplementary Fig. 9.

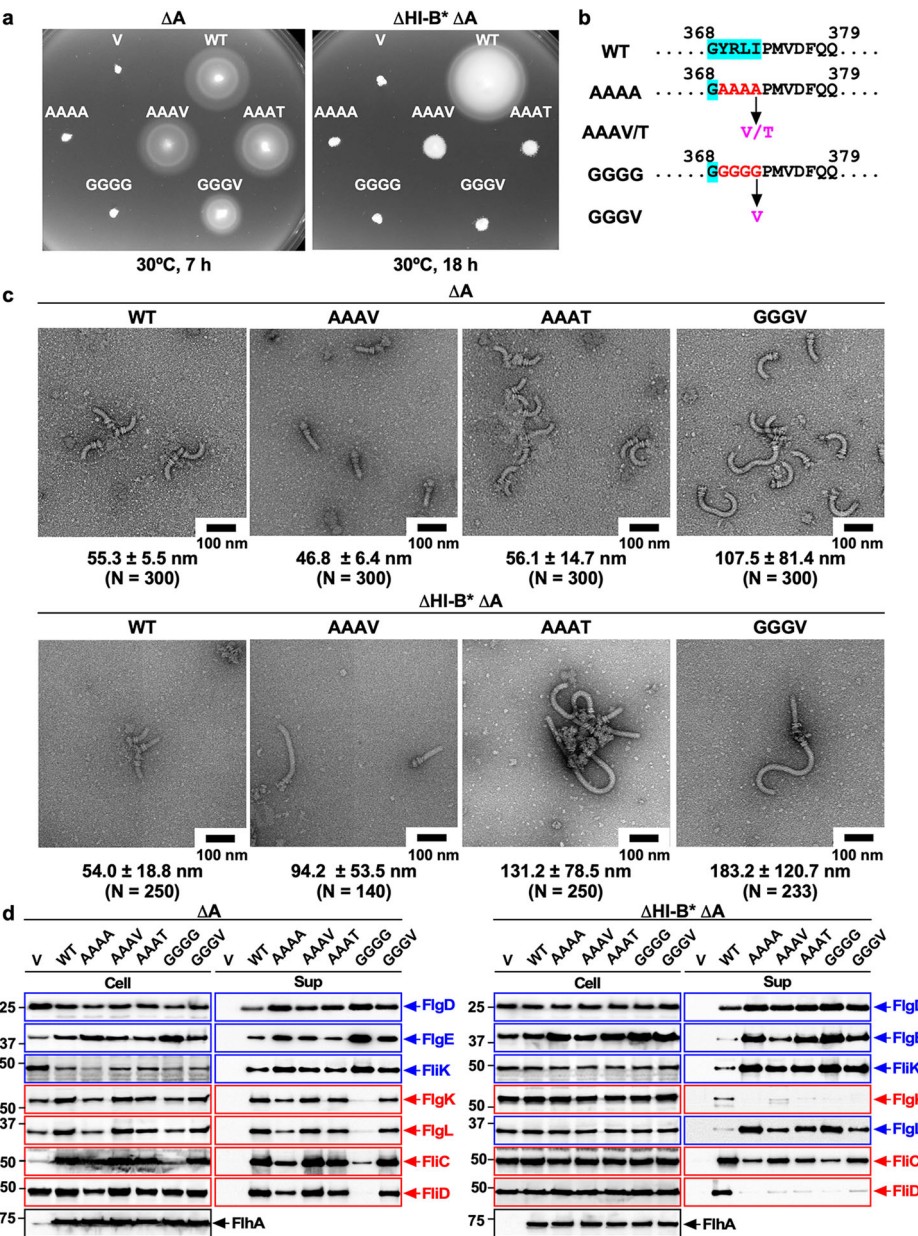

and 107.5 ± 81.4 nm (N = 300), respectively, which are much shorter than the polyhook lengths of their parental mutant strains and are much closer to 55.3 ± 5.5 nm (N = 300) of wild-type cells. Consistently, these suppressor mutations reduced the secretion levels of FlgD, FlgE, and FliK compared to the AAAA and GGGG mutants (Fig. 3d, left panel). FliK deletion caused complete inhibition in the motility of these three suppressor mutants (Supplementary Fig. 3a) by producing polyhooks without filament attached (Supplementary Fig. 3b). These results suggest that the fT3SS with the AAAV, AAAT, or GGGV mutation can receive the hook length signal from the FliK ruler protein and terminate the export of RH-type substrates at a more appropriate timing of hook assembly compared to their parental mutant strains.

To investigate whether these suppressor mutations restore the export of F-type substrates to the wild-type levels, we analyzed the secretion levels of FlgK, FlgL, FliC, and FliD (Fig. 3d, left panel). The amounts of FlgK, FlgL, FliC, and FliD secreted by the AAAV and AAAT suppressor mutants were greater than those seen in the AAAA mutant and similar to the wild-type levels. Consistently, they produced longer filaments than the AAAA mutant (Supplementary Fig. 1). The secretion levels of F-type substrates by the

GGGV suppressor mutant were also much higher than those by the GGGG mutant, but lower than the wild-type levels (Fig. 3d, left panel). Consistent with this, the GGGV suppressor mutant produced shorter filaments than wild-type cells (Supplementary Fig. 1). Because these F-type substrates require their cognate export chaperones for efficient docking to the FlhAC ring for export[28,29], these results suggest that, compared to FlhAC with the AAAA or GGGG mutation, FlhAC with the AAAV, AAAT, or GGGV mutation can make a more appropriate chaperone binding site in the conserved hydrophobic dimple of FlhAC once HBB assembly is complete. The conformational change of the GYXLI motif of FlhAC would be necessary not only for efficient transition of the FlhAC ring from the RH state to the F state but also for the formation of appropriate chaperone binding sites in the ring.

## Characterization of extragenic suppressor mutants isolated from the AAAA mutant

The extragenic suppressor mutations isolated from the AAAA mutant, *fliK(A405V)* and *fliK(Q338R)*, are located within FliKC, which directly binds to FlhBC and catalyzes substrate specificity switching of the fT3SS from the

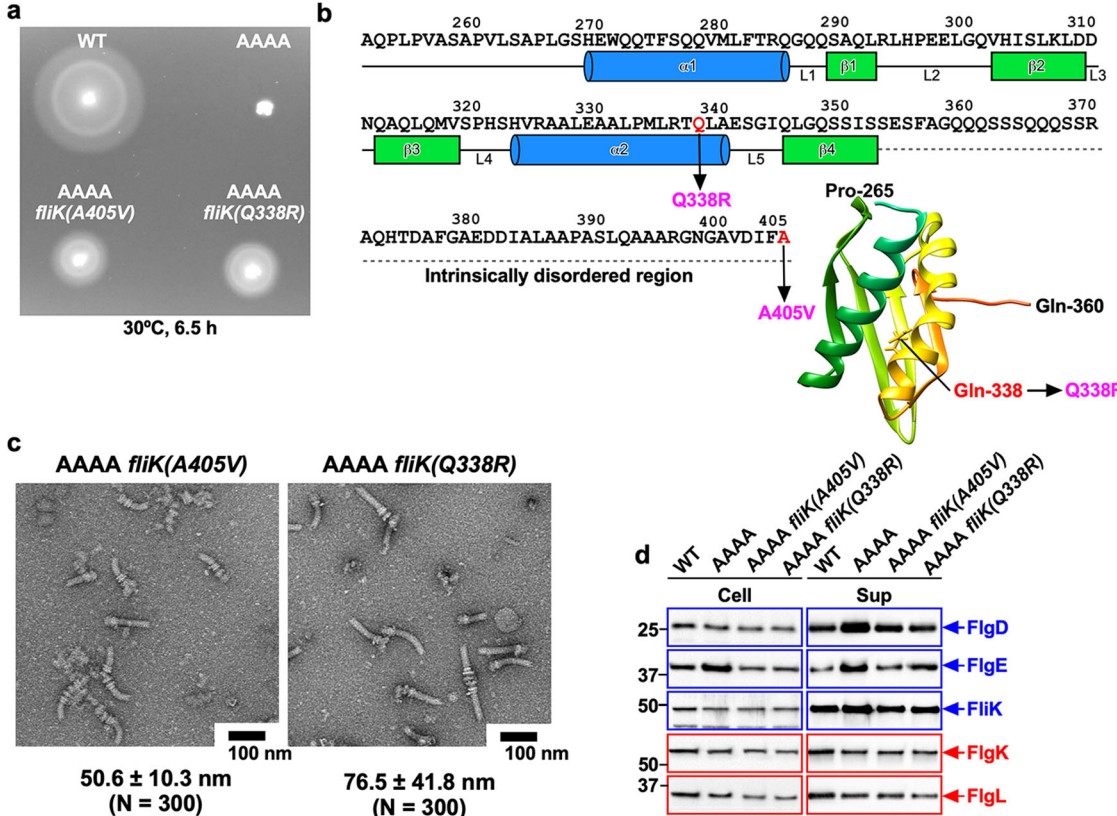

**Fig. 4 | Isolation of extragenic suppressor mutants from the AAAA mutant.**
**a** Motility of NH001 carrying pMM130 (WT), MMA130A4 (AAAA), MMA130A4-3 [AAAA *fliK(A405V)*], and MMA130A4-10 [AAAA *fliK(Q338R)*] in soft agar. Plates were incubated at 30 °C for 6.5 h. **b** Location of extragenic suppressor mutations in the C-terminal domain of FliK (FliK$_C$). The compactly folded core domain of FliK$_C$ (PDB ID: 2RRL) consists of residues 268–352 and is directly involved in substrate specificity switching of the fT3SS from the RH-type to the F-type. The secondary structures are shown below the amino acid sequence of FliK. Residues of 353–405 are intrinsically disordered in solution. The Cα backbone is color-coded from green to orange, going through the rainbow colors from the N-terminus to C-terminus. Extragenic suppressor mutations are highlighted in magenta. **c** Electron micrographs of hook-basal bodies isolated from the above strains. The average hook length and standard deviations are shown. N indices the number of hook-basal bodies that were measured. **d** Secretion assays of flagellar proteins. Immunoblot, using polyclonal anti-FlgD (1st row), anti-FlgE (2nd row), anti-FliK (3rd row), anti-FlgK (4th row) or anti-FlgL (5th row) antibody, of whole cell proteins (Cell) and culture supernatants (Sup) prepared from the above strains. RH-type and F-type substrates are highlighted in blue and red, respectively. Molecular mass markers (kDa) are shown on the left. The regions of interest were cropped from original immunoblots shown in Supplementary Fig. 10.

RH-type to the F-type (Fig. 4b). Therefore, we investigated whether these *fliK* mutations shorten the polyhook length of the AAAA mutant (Fig. 4c). The hook lengths of the AAAA *fliK(A405V)* and AAAA *fliK(Q338R)* mutants were 50.6 ± 10.3 nm (N = 300) and 76.5 ± 41.8 nm (N = 300), respectively, compared to 225.7 ± 176.8 nm (N = 300) of the AAAA mutant. Consistently, these two *fliK* mutations reduced the secretion levels of FlgD and FlgE (Fig. 4d). Because neither cytoplasmic nor secretion level of FliK was affected by these *fliK* mutations (Fig. 4d), the fT3SS with the AAAA mutation can receive the hook length signal more efficiently from FliK with the A405V or Q338R mutation than that from wild-type FliK to terminate hook assembly.

To investigate the export switching efficiency of these extragenic suppressor mutants, we analyzed the secretion levels of FlgK and FlgL (Fig. 4d). The levels of FlgK and FlgL secreted by the AAAA *fliK(A405V)* and AAAA *fliK(Q338R)* mutants were essentially the same as those seen in the AAAA mutant. Consistently, they produced short filaments in a way like the AAAA mutant (Supplementary Fig. 1). These observations suggest that the termination of RH-type protein export and activation of F-type protein export are independent processes that are not tightly coupled with each other.

To test whether these second site *fliK* mutations by themselves affect the export switching function of the fT3SS, we constructed strains containing only either *fliK(A405V)* or *fliK(Q338R)* mutation. Motility of the *fliK(A405V)* and *fliK(Q338R)* mutants were almost the same as that of wild-

type cells (Supplementary Fig. 4a). Because neither flagellar protein export by the fT3SS or hook length control was affected by these two *fliK* mutations (Supplementary Fig. 4b, c), we propose that these *fliK* mutations may induce a conformational change in FliK$_C$ to allow its stronger action on FlhB$_C$, whereby inducing a conformational change of the FlhA$_C$ ring with the AAAA mutation for more efficient termination of RH-type protein export.

**Effect of mutations in the GYXLI motif on FlhA$_C$ ring formation**
High-speed atomic force microscopy (HS-AFM) has shown that the interactions of FlhA$_{L-C}$ with the D1 and D3 domains of its closest FlhA$_C$ subunit (Fig. 1) are important for stable FlhA$_C$ ring formation in solution[22]. Because the W354A mutation in FlhA$_{L-C}$ not only inhibits FlhA$_C$ ring formation but also reduces the binding affinity of FlhA$_C$ for the FlgN-FlgK chaperone-substrate complex[22], the FlhA$_C$ ring observed by HS-AFM reflects an F-type ring structure. To obtain direct evidence that the GYXLI motif of FlhA is involved in the structural transition of FlhA$_C$ from the RH state to the F state, we purified the N-terminally His-tagged FlhA$_{C-AAAA}$ and FlhA$_{C-GGGG}$ monomers by size exclusion chromatography (Supplementary Fig. 5a) and analyzed their ring forming ability by HS-AFM (Fig. 6). FlhA$_{C-AAAA}$ formed the ring structure like wild-type FlhA$_C$ whereas FlhA$_{C-GGGG}$ did not, indicating that the GGGG mutation inhibits the interaction of FlhA$_{L-C}$ with its closest FlhA$_C$ subunit while the AAAA mutation does not. Because far-UV CD measurements revealed that the AAAA and GGGG mutations did not severely impair the entire FlhA$_C$

**Fig. 5 | Effect of the I372A and I372G substitutions in the conserved GYXLI motif of FlhA on flagellar protein export in the presence and absence of FliH and FliI. a** Motility of the *Salmonella* NH001 (Δ*flhA*, indicated as ΔA) or NH003 [Δ*fliH-fliI flhB*(P28T) Δ*flhA*, indicated as ΔHI-B* ΔA] strain transformed with pTrc99AFF4 (indicated as V), pMM130 (indicated as WT), pMKM130(I372A) (indicated as I372A), or pMKM130(I372G) (indicated as I372G) in the presence (left panel) and absence (right panel) of FliH and FliI. Soft tryptone agar plates were incubated at 30 °C for 6.5 hours (left panel) or 24 h (right panel). **b** Immunoblot, using polyclonal anti-FlgD (1st row), anti-FliK (2nd row), anti-FlgK (3rd row), anti-FlgL (4th row), anti-FliC 5th row) or anti-FlhA$_C$ (6th row) antibody, of whole cell proteins (Cell) and culture supernatants (Sup) prepared from the above transformants. RH-type and F-type substrates are highlighted in blue and red, respectively. Molecular mass markers (kDa) are shown on the left. The regions of interest were cropped from original immunoblots shown in Supplementary Fig. 11. **c** Electron micrographs of hook-basal bodies isolated from the *flhA(I372A)* and *flhA(I372G)* mutants. The average hook length and standard deviations are shown. N indices the number of hook-basal bodies that were measured.

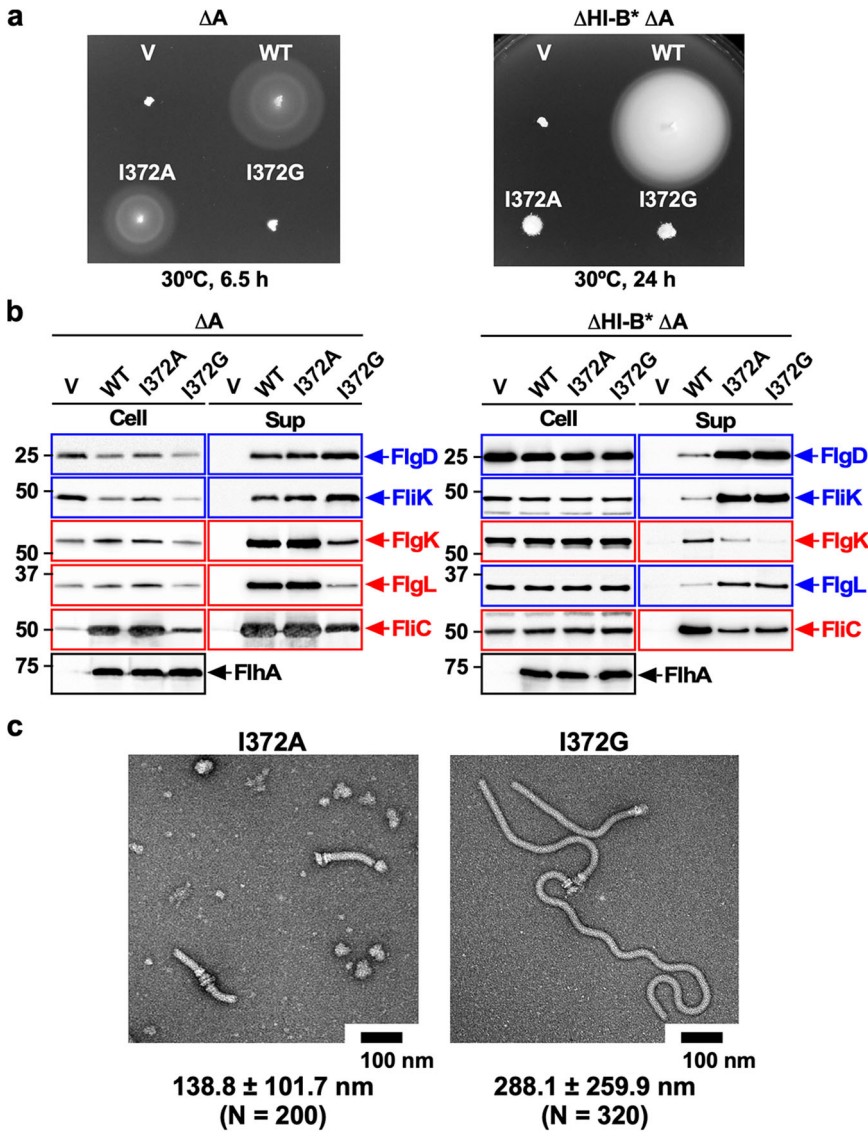

structure (Supplementary Fig. 5b), we suggest that an appropriate conformational change in the GYXLI motif is required for FlhA$_{L-C}$ to bind to the D1 and D3 domains of the closest subunit in the ring.

We next investigated whether the intragenic suppressor mutations affect FlhA$_C$ ring formation. FlhA$_{C-AAAV}$ formed the ring structure as FlhA$_{C-AAAA}$ but FlhA$_{C-GGGV}$ failed to form the ring structure as FlhA$_{C-GGGG}$. This was rather surprising because the fT3SS with FlhA$_{C-GGGV}$ can switch its substrate specificity from the RH-type to the F-type to a significant degree (Fig. 3). So, we hypothesized that the GGGG and GGGV mutations both lock FlhA$_C$ in the RH state but FlhA$_C$ with the GGGV mutation may be able to make the transition from the RH state to the F state with a support of other proteins when it receives the hook length signal from FliK.

**Effect of mutations in the GYXLI motif on the interaction of FlhA$_C$ with the FlgN-FlgK complex**
Flagellar export chaperone in complex with their cognate F-type proteins bind to a hydrophobic dimple located at an interface between domains D1 and D2 of FlhA$_C$, allowing the fT3SS to efficiently transport the F-type proteins to the distal end of the growing flagellar structure[28–30]. Because the AAAA mutation reduced the secretion levels of F-type proteins whereas its intragenic AAAV suppressor mutation restored those secretion levels to the wild-type levels (Fig. 3d, left panel), we investigated whether these mutations

affect the interaction of FlhA$_C$ with the FlgN-FlgK chaperone-substrate complex by GST affinity chromatography. Unlike wild-type FlhA$_C$, only a very small amount of FlhA$_{C-AAAA}$ co-purified with the GST-FlgN-FlgK complex, and its intragenic suppressor mutation increased the binding affinity for the FlgN-FlgK complex although not to the wild-type level (Fig. 7a). Because neither the AAAA nor AAAV mutation inhibited FlhA$_C$ ring formation (Fig. 6), a proper conformational change of the GYXLI motif of FlhA$_C$ would also be required for the formation of an appropriate chaperone binding site in the hydrophobic dimple of FlhA$_C$.

Because the GGGG and GGGV mutations both inhibited FlhA$_C$ ring formation (Fig. 6), we investigated whether they also prevent the FlgN-FlgK complex from binding to FlhA$_C$. FlhA$_C$ with the GGGG or GGGV mutation did not co-purified with the GST-FlgN-FlgK complex at all (Fig. 7a). Therefore, the GGGG and GGGV mutations might stabilize the FlhA$_C$ conformation in the RH state, thereby inhibiting the interaction of FlhA$_C$ with the FlgN-FlgK complex in vitro.

**Effect of mutations in the GYXLI motif on the interaction of FlhA$_C$ with FliJ**
It has been proposed that an interaction between FlhA$_C$ and FliJ may be required for efficient transition of the FlhA$_C$ ring from the RH state to the F state[22,31]. Therefore, we analyzed the FlhA$_C$-FliJ interaction by GST affinity chromatography. FlhA$_C$ with the AAAA, AAAV, GGGG, or GGGV

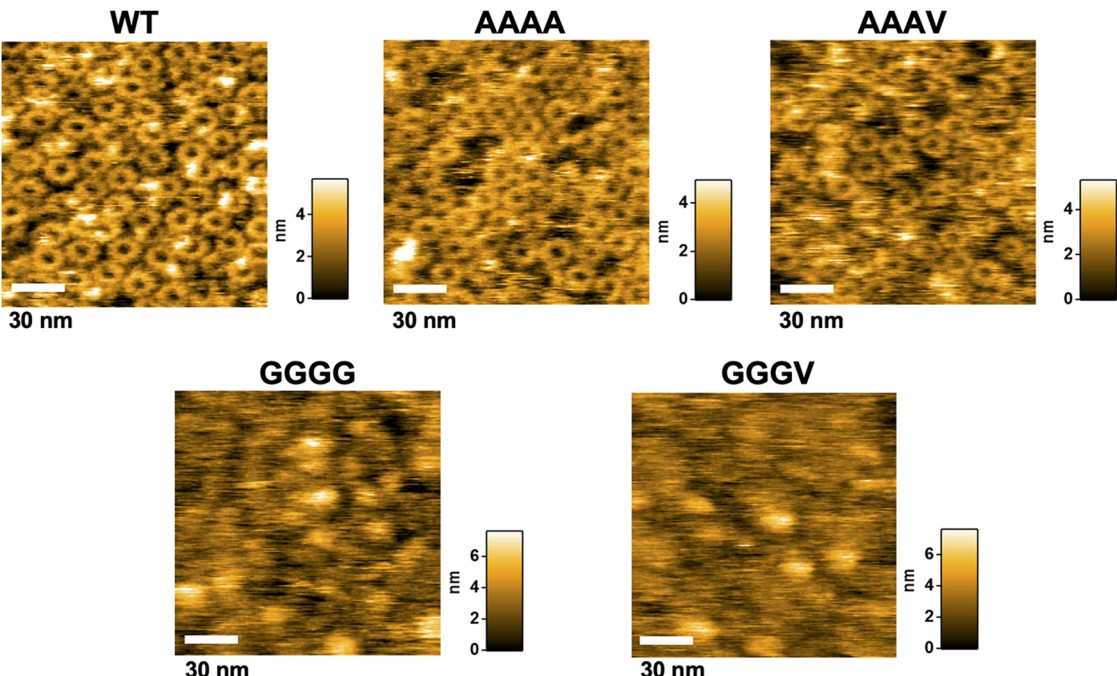

**Fig. 6 | Effect of mutations in the GYXLI motif on FlhA$_C$ ring formation.** Typical HS-AFM images of His-FlhA$_C$ (WT) and its mutant variants with either AAAA, AAAV, GGGG, or GGGV mutation placed on mica surface in a buffer at a protein concentration of 2 μM. All images were recorded at 200 ms/frame in a scanning area of 100 × 100 nm$^2$ with 150 × 150 pixels. Color bar on the right of each image indicates a range of particle heights (nm).

**Fig. 7 | Effect of mutations in the GYXLI motif on the interaction of FlhA$_C$ with the FlgN-FlgK chaperone-substrate complex and FliJ.** Mixtures (L) of purified His-FlhA$_C$ (WT, 1st row), His-FlhA$_{C-AAAA}$ (AAAA, 2nd row), His-FlhA$_{C-AAAV}$ (AAAV, 3rd row), His-FlhA$_{C-GGGG}$ (GGGG, 4th row), or His-FlhA$_{C-GGGV}$ (GGGV, 5th row) with GST-FlgN in complex with FlgK (**a**) or GST-FliJ (**b**) were dialyzed overnight against PBS, followed by GST affinity chromatography. Flow through fraction (F.T.), wash fractions (W) and elution fractions (E) were analyzed by Coomassie Brilliant blue staining. Molecular mass markers (kDa) are shown on the left. The regions of interest were cropped from original CBB-stained gel images shown in Supplementary Fig. 12.

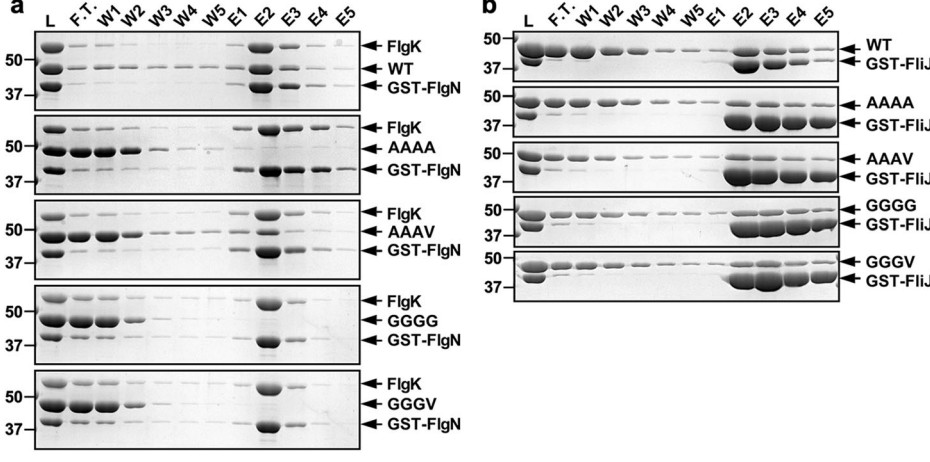

mutation co-purified with GST-FliJ as wild-type FlhA$_C$ (Fig. 7b), indicating that these mutations do not reduce the binding affinity of FlhA$_C$ for FliJ. Therefore, the GYXLI motif of FlhA$_C$ is not involved in the interaction with FliJ.

**Effect of removal of both FliH and FliI on the export switching function of FlhA with the AAAV, AAAT or GGGV mutation**

We found that the AAAV and GGGV mutations both reduced the binding affinity of FlhA$_C$ for the FlgN-FlgK complex (Fig. 7a). Because FlhA requires the support of FliH and FliI to efficiently exert its export function[36–38], we analyzed the effect of the AAAV, AAAT, or GGGV mutation on the export switching function of FlhA in the Δ*fliH-fliI flhB(P28T)* (hereafter referred to as ΔHI-B*) mutant background. Neither flagella-driven motility in soft agar nor flagellar protein export by the fT3SS was affected by the *flhB(P28T)* (hereafter referred to as B*) mutation alone (Supplementary Fig. 6). Unlike in the presence of FliH and FliI, the motility of the ΔHI-B* AAAV, ΔHI-B* AAAT, and ΔHI-B* GGGV mutants was worse than that of the ΔHI-B*

mutant (Fig. 3a, right panel). The amounts of FlgD, FlgE, and FliK secreted from the ΔHI-B* AAAV, ΔHI-B* AAAT, and ΔHI-B* GGGV mutants were higher than those of the ΔHI-B* mutant whereas the secretion levels of FlgK, FliC, and FliD were lower in these three mutants than in the ΔHI-B* strain (Fig. 3d, right panel), suggesting that these three mutations affect the hook length significantly in the absence of FliH and FliI. To confirm this, we analyzed their hook length. The hook length of the ΔHI-B* strain was 54.0 ± 18.8 nm ($N = 250$), showing a much broader length distribution compared to the wild-type, in agreement with a previous report[36]. The hook lengths of the ΔHI-B* AAAV, ΔHI-B* AAAT, and ΔHI-B* GGGV mutants were 94.2 ± 53.5 nm ($N = 140$), 131.2 ± 78.5 nm ($N = 250$), and 183.2 ± 120.7 nm ($N = 233$), respectively. Furthermore, the *flhA(I372A)* and *flhA(I372G)* mutations inhibited substrate specificity switching of the fT3SS from the RH-type to the F-type in the absence of FliH and FliI, thereby inhibiting the motility of the ΔHI-B* mutant (Fig. 5b, right panels). These results suggest that the FlhA$_C$ ring requires the support of FliH and FliI to efficiently undergo a structural transition from the RH state to the F state.

## Effect of the AAAA and GGGG mutations on FlgL secretion in the presence and absence of FliH and FliI

Although FlgK and FlgL belong to the same F-type class and require FlgN chaperone, we found a peculiar change in the secretion level of FlgL by the presence and absence of FliH and FliI. In the presence of FliH and FliI, the amount of FlgL secreted from the AAAA, GGGG, B* AAAA, and B* GGGG mutants was less than those of the wild-type strain, as was the case for other F-type proteins (Fig. 3d, left panel and Supplementary Fig 6b). However, in the absence of FliH and FliI, the amount of FlgL secreted from the B* AAAA and B* GGGG mutants was higher than those secreted from the wild-type, as was the case for the RH-type proteins (Fig. 3d, right panel). Because the AAAA and GGGG mutations inhibit substrate specificity switching of the fT3SS from the RH-type to the F-type even in the presence of FliH and FliI, thereby producing polyhooks (Fig. 2), we suggest that the fT3SS with the AAAA or GGGG mutation recognizes FlgL as an RH-type substrate rather than an F-type substrate in the absence of FliH and FliI whereas it recognizes FlgL properly as an F-type substrate in the presence of FliH and FliI.

Both FlgK and FlgL require FlgN for efficient binding to FlhA$_C$, allowing these two proteins to be efficiently transported by the fT3SS. So, when the FlgN-FlhA$_C$ interaction is impaired, the secretion levels of FlgK and FlgL are significantly reduced[28,29]. To investigate whether FlgL secretion by the ΔHI-B* AAAA and ΔHI-B* GGGG mutants is dependent on FlgN, we introduced a ΔflgN::tetRA allele into the ΔHI-B* ΔA, ΔHI-B*, ΔHI-B* AAAA and ΔHI-B* GGGG mutants and analyzed the secretion level of FlgL. FlgN has been reported to be essential for the export of both RH-type and F-type substrates because it also acts as an activator of the transmembrane export gate complex of the fT3SS when the cytoplasmic ATPase complex is dysfunctional[10]. Therefore, we also measured the secretion level of FlgD. As expected, neither FlgD nor FlgL was secreted from the ΔHI-B* cells containing the ΔflgN::tetRA allele (Fig. 8). The AAAA and GGGG mutations overcame the effect of FlgN deletion on flagellar protein export, thereby allowing both FlgD and FlgL to be secreted extracellularly even in the absence of FliH and FliI (Fig. 8). This indicates that the fT3SS with the AAAA or GGGG mutation does not require FlgN for FlgL secretion in the absence of FliH and FliI. The flhA(D456V) and flhA(T490M) mutations in the conserved hydrophobic dimple of FlhA$_C$ have been shown to be able to bypass the FlgN defect to a significant degree[10,28]. Because the flhA(G368C) mutation in the GYXLI motif affects a conformation of the hydrophobic dimple[33,34], the AAAA and GGGG mutations may induce a required conformational change in the conserved dimple of FlhA$_C$ to allow the transmembrane export gate complex to become an active protein transporter.

## Discussion

The highly conserved GYXLI motif of FlhA$_C$ acts as a structural switch to facilitate cyclic domain motions of FlhA$_C$ through periodically remodeling its hydrophobic side-chain interaction networks[34]. The flhA(G368C) mutation in the GYXLI motif affects substrate specificity switching of the fT3SS from the RH-type to the F-type in the absence of FliH and FliI. Furthermore, this mutation reduces the binding affinity of FlhA$_C$ for export chaperones in complex with their cognate F-type substrates[33,34]. These observations raised the possibility that the GYXLI motif is also involved in substrate specificity switching of the fT3SS from the RH-type to the F-type. Therefore, we analyzed the effect of the AAAA and GGGG mutations in the GYXLI motif on the export switching function of the fT3SS. The AAAA mutation delayed substrate specificity switching of the fT3SS from the RH-type to the F-type, resulting in a polyhook-filament phenotype, while the GGGG mutation totally inhibited the substrate specificity switching, causing a polyhook phenotype (Fig. 2). The intragenic AAAV/T and GGGV suppressor mutations restored the export switching function of the fT3SS to a significant degree (Fig. 3). Because conformational changes in the GYXLI motif of FlhA$_C$ cause a reorganization of the hydrophobic side-chain interaction network throughout the entire FlhA$_C$ structure, allowing FlhA$_C$ to take different conformations[34], a specific conformational change of the GYXLI motif would be necessary for the efficient and robust structural

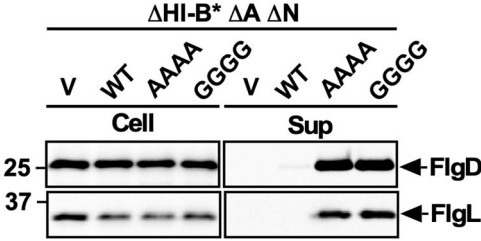

**Fig. 8 | Effect of FlgN deletion on flagellar protein export by the AAAA and GGGG mutants in the absence of FliH and FliI.** Immunoblot, using polyclonal anti-FlgD (1st row) and anti-FlgL (2nd row), of whole cell proteins and culture supernatant fractions prepared from the *Salmonella* NH003gN [Δ*fliH-fliI flhB(P28T)* Δ*flhA* Δ*flgN::tetRA*, indicated as ΔHI-B* ΔA ΔN] strain transformed with pTrc99AFF4 (V), pMM130 (WT), pMKM130-A4 (AAAA), or pMKM130-G4 (GGGG). The positions of molecular mass markers are indicated on the left. The regions of interest were cropped from original immunoblots shown in Supplementary Fig. 13.

transition of the FlhA$_C$ ring from the RH state to the F state upon hook completion.

Residues 301–350 of the FliK$_C$ core domain and the last five residues of the intrinsically disordered FliK$_C$ region are important for the export switching function of FliK (Fig. 4b)[39]. Photo-crosslinking experiments have shown a direct interaction between the FliK$_C$ core domain and FlhB$_C$[23]. The flhB(P270A) mutation in FlhB$_C$ inhibits substrate specificity switching of the fT3SS from the RH-type to the F-type at an appropriate timing of hook assembly, thereby producing polyhooks with or without filament attached[40]. This mutation, however, does not affect the FliK$_C$-FlhB$_C$ interaction, indicating that FliK cannot efficiently transmit the hook length signal to the FlhA$_C$ ring through the interaction between FliK and FlhB$_{C-P270A}$[24]. The flhA(A489E) suppressor mutation, located at the chaperone binding site of FlhA$_C$, increases the probability of filament formation, thereby improving the motility of the flhB(P270A) mutant. This suggests that the interaction between FlhB$_C$ and FlhA$_C$ is critical for the initiation of F-type protein export[24]. Here, we showed that the fliK(Q338R) and fliK(A405V) mutations shortened the length of polyhooks produced by the AAAA mutant, thereby improving the motility in soft agar (Fig. 4). The fliK(Q338R) and fliK(A405V) mutations alone showed no significant motility phenotype (Supplementary Fig. 4), suggesting that they do not inhibit or facilitate the interaction of FliK$_C$ with FlhB$_C$. Therefore, we propose that the fliK(Q338R) and fliK(A405V) mutations allow the FliK$_C$-FlhB$_C$ complex to efficiently bind to FlhA$_C$ with the AAAA mutation, thereby inducing the dissociation of FlhA$_{L-C}$ from the conserved hydrophobic dimple to terminate RH-type protein export at a more appropriate timing of hook assembly. However, these extragenic suppressor mutations in the fliK gene did not improve the efficiency of F-type protein export at all (Fig. 4d), and these suppressor mutants produced short filaments like the AAAA mutant (Supplementary Fig. 1). Thus, it seems unlikely that the FliK$_C$-FlhB$_C$ complex is directly involved in the formation of the appropriate chaperone-binding site in the hydrophobic dimple of FlhA$_C$ after complete cessation of RH-type protein export.

The hook length of *Salmonella* is controlled to about 55 nm with an error of about 10%. The average hook length of the ΔHI B* mutant is nearly the same as that of the wild-type strain, but the hook length distribution of this mutant is much broader than that of the wild type (Fig. 3c)[36]. Therefore, FliH and FliI are necessary for FliK to measure hook length in a more accurate manner. Here, we showed that removal of both FliH and FliI from the intragenic AAAV/T and GGGV suppressor mutants markedly reduce the substrate specificity switching efficiency of the fT3SS, resulting in nearly twice longer polyhooks (Fig. 3c). Because FlhA$_{L-C}$ binds tightly to the hydrophobic dimple of FlhA$_C$ during HBB assembly[31], biological energy may be required for efficient dissociation of FlhA$_{L-C}$ from the dimple. FliH, FliI, and FliJ assemble into the cytoplasmic ATPase ring complex at the base of the flagellum, and ATP hydrolysis by the FliI ATPase turns an inactive

**Table 1 | Strains and plasmids used in this study**

| Strain/Plasmid | Relevant characteristics | References |
|---|---|---|
| **E. coli** | | |
| BL21 Star (DE3) | Overexpression of proteins | Novagen |
| **Salmonella** | | |
| SJW1103 | Wild-type for motility and chemotaxis | [52] |
| SJW1368 | ΔcheW–flhD | [53] |
| NH001 | ΔflhA | [37] |
| NH002 | flhB(P28T) ΔflhA | [37] |
| NH003 | ΔfliH-fliI flhB(P28T) ΔflhA | [37] |
| TH8426 | ΔfliK | [39] |
| NH001iK | ΔflhA ΔfliK::tetRA | This study |
| NH003gN | ΔfliH-fliI flhB(P28T) ΔflhA ΔflgN::tetRA | This study |
| MMA130A4 | NH001 harboring pMKM130-A4 | This study |
| MMA130A4-3 | NH001 harboring pMKM130-A4 fliK(A405V) | This study |
| MMA130A4-5 | NH001 harboring pMKM130-A3V | This study |
| MMA130A4-7 | NH001 harboring pMKM130-A3T | This study |
| MMA130A4-10 | NH001 harboring pMKM130-A4 fliK(Q338R) | This study |
| MMA130G4 | NH001 harboring pMKM130-G4 | This study |
| MMA130G4-3 | NH001 harboring pMKM130-G3V | This study |
| MMK130-3 | fliK(A405V) | This study |
| MMK130-10 | fliK(Q338R) | This study |
| **Plasmids** | | |
| pTrc99AFF4 | Modified pTrc expression vector | [54] |
| pGEX-6p-1 | Expression vector | GE Healthcare |
| pMKGK2 | pTrc99A/ FlgK | [49] |
| pMM104 | pET19b/ His-FlhA_C (residues 211–692) | [41] |
| pMM130 | pTrc99AFF4/ FlhA | [55] |
| pMKM130-A4 | pTrc99AFF4/ FlhA(T369A/R370A/L371A/I372A) | [34] |
| pMKM130-G4 | pTrc99AFF4/ FlhA(T369G/R370G/L371G/I372G) | [34] |
| pMMGN101 | pGEX-6p-1/ GST-FlgN | [28] |
| pMMJ1001 | pGEX-6p-1/ GST-FliJ | [35] |
| pMKM104-A4 | pET19b/ His-FlhA_C(T369A/R370A/L371A/I372A) | This study |
| pMKM104-A3V | pET19b/ His-FlhA_C(T369A/R370A/L371A/I372V) | This study |
| pMKM104-G4 | pET19b/ His-FlhA_C(T369G/R370G/L371G/I372G) | This study |
| pMKM104-G3V | pET19b/ His-FlhA_C(T369G/R370G/L371G/I372V) | This study |
| pMKM130-A3V | pTrc99AFF4/ FlhA(T369A/R370A/L371A/I372V) | This study |
| pMKM130-A3T | pTrc99AFF4/ FlhA(T369A/R370A/L371A/I372T) | This study |
| pMKM130-G3V | pTrc99AFF4/ FlhA(T369G/R370G/L371G/I372V) | This study |
| pMKM130(I372A) | pTrc99AFF4/ FlhA(I372A) | This study |
| pMKM130(I372G) | pTrc99AFF4/ FlhA(I372G) | This study |

export gate complex into a highly active protein transporter through the interaction of FliJ with FlhA_L[3]. The W354A and E351A/D356A mutations in FlhA_{L-C} inhibit the export of F-type proteins but not that of RH-type proteins. Because these two mutations also reduce the binding affinity of FlhA_C for FliJ[22,31], it has been proposed that the interaction between FlhA_L and FliJ may also be required to efficiently switch the FlhA_C ring structure from the RH state to the F state upon hook completion[22]. Because the

AAAV/T and GGGV mutations did not inhibit the interaction of FlhA_C with FliJ (Fig. 7b), we propose that the efficient transition of the FlhA_C ring from the RH state to the F state induced by the FliJ-FlhA_L interaction may require energy derived from ATP hydrolysis by the cytoplasmic ATPase ring complex.

The highly conserved Tyr-106 residue of FliT is required for the interaction with FlhA_C (PDB ID: 6CH2)[29,30]. Comparison of the FlhA_C structures with and without FliT bound has shown that the binding of FliT to FlhA_C induces a rotation of domain D2 relative to domain D1 of FlhA_C through a conformational change in the GYXLI motif, thereby allowing Tyr-106 of FliT to bind efficiently to the hydrophobic dimple of FlhA_C (Supplementary Fig. 7). Purified FlhA_C with the AAAV mutation formed the nonameric ring like wild-type FlhA_C (Fig. 6), suggesting that FlhA_C with the AAAV mutation prefers to adopt an F-type conformation in solution. However, this AAAV mutation reduced the binding affinity of FlhA_C for the FlgN-FlgK complex (Fig. 7a), suggesting that this mutation affects the rotation of domain D2 relative to domain D1. The AAAV mutation inhibited the secretion of FlgK and FliD in the absence of FliH and FliI but not in their presence (Fig. 3d). Therefore, we suggest that FlhA_C requires the support of FliH and FliI to maintain an appropriate conformation of the GYXLI motif to facilitate efficient docking of the chaperone-substrate complex to FlhA_C.

The RH-type substrates also bind to FlhA_C[33,41], and FliH and FliI are required for hierarchical targeting of export substrates and chaperone-substrate complexes to FlhA_C[36]. Here, we found that the fT3SS with the AAAA or GGGG mutation in FlhA_C recognizes FlgL as an F-type substrate in the presence of FliH and FliI but as an RH-type substrate in their absence (Fig. 3d). This is also true for the flhA(I372A) and flhA(I372G) mutations (Fig. 5b, right panel). These observations indicate that FliH and FliI help the fT3SS correct substrate recognition errors by the AAAA, GGGG, flhA(I372A), or flhA(I372G) mutation that occur during flagellar assembly. Because FlhA_C can take different conformations through conformational changes in the GYXLI motif[54], we propose that FliH and FliI also support FlhA in taking appropriate conformations at different steps of flagellar protein export to bring strict order in the export substrates for efficient assembly of the flagellum and that a specific conformational change of the GYXLI motif is required for this FlhA function to be properly performed. The RH-type substrates have a common hydrophobic sequence (FXXXΦ in which Φ is a hydrophobic residue) in their N-terminal region, named gate recognition motif (GRM) responsible for an interaction with FlhB_C. The interaction between the GRM and FlhB_C is essential for RH-type protein export[42–44]. Because this GRM sequence is not present in the N-terminal region of FlgL, FlhA_C may also recognize something else as the RH-type substrate signal for FlgL export in the absence of FliH and FliI.

## Methods

### Bacterial strains, plasmids, P22-mediated transduction, and media

*Salmonella* strains and plasmids used in this study are listed in Table 1. To identify and purify extragenic suppressor mutations, P22-mediated transduction was performed using P22HT*int*[45]. L-broth contained 10 g of Bacto-Tryptone, 5 g of yeast extract and 5 g of NaCl per liter. Soft tryptone agar plates contained 10 g of Bacto Tryptone, 5 g of NaCl and 3.5 g of Bacto-Agar per liter. Ampicillin and tetracycline were added as needed at a final concentration of 100 μg ml$^{-1}$ and 15 μg ml$^{-1}$, respectively.

### DNA manipulations

DNA manipulations were performed using standard protocols. Site-directed mutagenesis was carried out using Prime STAR Max Premix as described in the manufacturer's instructions (Takara Bio). All mutations were confirmed by DNA sequencing (Eurofins Genomics).

### Motility assays in soft agar

Fresh colonies were inoculated onto soft tryptone agar plates and incubated at 30 °C. At least six measurements were carried out.

## Secretion assays

*Salmonella* cells were grown in 5 ml of L-broth containing ampicillin with shaking until the cell density had reached an $OD_{600}$ of ca. 1.2–1.4. Cultures were centrifuged to obtain cell pellets and culture supernatants, separately. The cell pellets were resuspended in sodium dodecyl sulfate (SDS)-loading buffer solution [62.5 mM Tris-HCl, pH 6.8, 2% (w/v) SDS, 10% (w/v) glycerol, 0.001% (w/v) bromophenol blue] containing 1 μl of 2-mercaptoethanol. Proteins in each culture supernatant were precipitated by 10% trichloroacetic acid and suspended in a Tris/SDS loading buffer (one volume of 1 M Tris, nine volumes of 1 X SDS-loading buffer solution)[46] containing 1 μl of 2-mercaptoethanol. Both whole cellular proteins and culture supernatants were normalized to a cell density of each culture to give a constant number of *Salmonella* cells. After boiling at 95 °C for 3 min, these protein samples were separated by SDS–polyacrylamide gel (normally 12.5% acrylamide) electrophoresis and transferred to nitrocellulose membranes (Cytiva) using a transblotting apparatus (Hoefer). Then, immunoblotting with polyclonal anti-FlgD, anti-FlgE, anti-FliK, anti-FlgK, anti-FlgL, anti-FlgM, anti-FliC, anti-FliD, or anti-$FlhA_C$ antibody as the primary antibody and anti-rabbit IgG, HRP-linked whole Ab Donkey (GE Healthcare) as the secondary antibody was carried out. Detection was performed with an ECL prime immunoblotting detection kit (GE Healthcare). Chemiluminescence signals were detected by a Luminoimage analyzer LAS-3000 (GE Healthcare). Bands of prestained protein molecular weight markers (Bio-Rad) transferred to each membrane were also photographed with the LAS-3000 under brightfield illumination and combined with each immunoblot image to identify the band of interest. All image data were processed with Photoshop (Adobe). At least three measurements were performed.

## Preparations of hook-basal bodies

*Salmonella* cells were grown in 500 ml of L-broth containing ampicillin at 30 °C with shaking until the cell density had reached an $OD_{600}$ of ca. 1.0. After centrifugation (10,000 g, 10 min, 4 °C), the cells were suspended in 20 ml of ice-cold 0.1 M Tris-HCl pH 8.0, 0.5 M sucrose, and EDTA and lysozyme were added at the final concentrations of 10 mM and 0.1 mg ml$^{-1}$, respectively. The cell suspensions were stirred for 30 min at 4 °C, and Triton X-100 and $MgSO_4$ were added at final concentrations of 1% (w/v) and 10 mM, respectively. After stirring on ice for 1 hour, the cell lysates were adjusted to pH 10.5 with 5 M NaOH and then centrifuged (10,000 g, 20 min, 4 °C) to remove cell debris. After ultracentrifugation (45,000 g, 60 min, 4 °C), pellets were resuspended in 10 mM Tris-HCl, pH 8.0, 5 mM EDTA, 1% Triton X-100, and this solution was loaded a 20–50% (w/w) sucrose density gradient in 10 mM Tris-HCl, pH 8.0, 5 mM EDTA, 1% Triton X-100. After ultracentrifugation (49,100 g, 13 h, 4 °C), hook-basal bodies and polyhook-basal bodies with or without filament attached were collected and ultracentrifuged (60,000 g, 60 min, 4 °C). Pellets were suspended in 50 mM glycine, pH 2.5, 0.1% Triton X100 to depolymerize the filaments. After ultracentrifugation (60,000 g, 60 min, 4 °C), pellets were resuspended in 50 μl of 10 mM Tris-HCl, pH 8.0, 5 mM EDTA, 0.1% Triton X100. Samples were negatively stained with 2% (w/v) uranyl acetate. Electron micrographs were taken using JEM-1400Flash (JEOL, Tokyo, Japan) operated at 100 kV. The length of hooks and polyhooks was measured by ImageJ version 1.52 (National Institutes of Health).

## Fluorescence microscopy

*Salmonella* cells were grown in 5 ml of L-broth containing ampicillin. The cells were attached to a cover slip (Matsunami glass, Japan), and unattached cells were washed away with motility buffer (10 mM potassium phosphate pH 7.0, 0.1 mM EDTA, 10 mM L-sodium lactate). Then, flagellar filaments were labelled using anti-FliC antibody and anti-rabbit IgG conjugated with Alexa Fluor 594 (Invitrogen). After washing twice with the motility buffer, the cells were observed by an inverted fluorescence microscope (IX-83, Olympus) with a 150× oil immersion objective lens (UApo150XOTIRFM, NA 1.45, Olympus) and an Electron-Multiplying Charge-Coupled Device

camera (iXon$^{EM}$+897-BI, Andor Technology). Fluorescence images of filaments labeled with Alexa Fluor 594 were merged with bright field images of cell bodies using ImageJ software version 1.52 (National Institutes of Health).

## Purification of His-tagged wild-type $FlhA_C$ and its mutant variants

His-$FlhA_C$ and its mutant variants were expressed in the *E. coli* BL21 Star (DE3) strain and purified by Ni-NTA affinity chromatography with a Ni-NTA agarose column (QIAGEN), followed by size exclusion chromatography (SEC) with a HiLoad 26/600 Superdex 75 pg column (GE Healthcare) at a flow rate of 2.5 ml min$^{-1}$ equilibrated with 50 mM Tris-HCl, pH 8.0, 100 mM NaCl, 1 mM EDTA.

## Analytical size exclusion chromatography

Analytical (SEC) was performed with a Superdex 75 HR 10/30 column (GE Healthcare)[47]. A 500 μl solution of purified His-$FlhA_C$ and its mutant variants (10 μM) were run on the SEC column equilibrated with 50 mM Tris-HCl, pH 8.0, 150 mM NaCl at a flow rate of 0.5 ml min$^{-1}$. Bovine serum albumin (66.4 kDa) and ovalbumin (44 kDa) were used as size markers. All fractions were run on SDS-PAGE and then analyzed by Coomassie Brilliant blue (CBB) staining.

## Far-UV CD spectroscopy

Far-UV CD spectroscopy of His-$FlhA_C$ or its mutant variants was carried out at room temperature using a Jasco-720 spectropolarimeter (JASCO International Co., Tokyo, Japan). The CD spectra of His-$FlhA_C$ and its mutant forms were measured in 20 mM Tris-HCl, pH 8.0 using a cylindrical fused quartz cell with a path length of 0.1 cm in a wavelength range of 200 nm to 260 nm[48]. Spectra were obtained by averaging five successive accumulations with a wavelength step of 0.5 nm at a rate of 20 nm min$^{-1}$, response time of 8 s, and bandwidth of 2.0 nm.

## Pull-down assays by GST affinity chromatography

FlgK was overexpressed in the *E. coli* BL21 Star (DE3) cells and purified by anion exchange chromatography with a Q-Sepharose high-performance column (GE Healthcare)[49]. GST-FlgN and GST-FliJ were purified by GST affinity chromatography with a glutathione Sepharose 4B column (GE Healthcare) from the soluble fractions from the *Salmonella* SJW1368 strain transformed with a pGEX-6p-1-based plasmid. Purified His-$FlhA_C$ or its mutant variants were mixed purified GST-FlgN/FlgK complex or purified GST-FliJ, and each mixture was dialyzed overnight against PBS (8 g of NaCl, 0.2 g of KCl, 3.63 g of $Na_2HPO_4$•$12H_2O$, 0.24 g of $KH_2PO_4$, pH 7.4 per liter) at 4 °C with three changes of PBS. A 5 ml solution of each mixture was loaded onto a glutathione Sepharose 4B column (bed volume, 1 ml) pre-equilibrated with 20 ml of PBS. After washing of the column with 10 ml PBS at a flow rate of ca. 0.5 ml min$^{-1}$, bound proteins were eluted with 50 mM Tris-HCl, pH 8.0, 10 mM reduced glutathione. Fractions were analyzed by SDS-PAGE with CBB staining.

## High-speed atomic force microscopy

HS-AFM imaging of $FlhA_C$ ring formation was carried out in solution using a laboratory-built HS-AFM[50,51]. A 2 μl solution of purified $FlhA_C$ monomer (2 μM) in 50 mM Tris-HCl, pH 8.0, 100 mM NaCl was placed on a freshly cleaved mica surface attached to a cylindrical glass stage. After incubating at room temperature for 3 min, the mica surface was rinsed thoroughly with 20 μl of 10 mM Tris-HCl, pH 6.8 to remove any residual molecules. Subsequently, the sample was immersed in a liquid cell containing 60 μl of 10 mM Tris-HCl, pH 6.8. AFM imaging was performed in a tapping mode using small cantilevers (AC7, Olympus). The AFM images were recorded at 200 ms/frame in a scanning area of $100 \times 100$ nm$^2$ with $150 \times 150$ pixels. For AFM image analysis, a low-pass filter was applied to remove random noise from the HS-AFM image, and a flattening filter was used to flatten the entire xy plane. Such image processing was carried out using a laboratory-developed software built on Igor Pro (HULINKS).

## Statistics and reproducibility

Statistical tests, sample size, and number of biological replicates are reported in the figure legends. Statistical analyses were done using Excel (Microsoft).

## Reporting summary

Further information on research design is available in the Nature Portfolio Reporting Summary linked to this article.

## Data availability

All data generated during this study are included in this published article, its Supplementary Information, and its Supplemental Data 1. Strains, plasmids, polyclonal antibodies, and all other data are available from the corresponding author upon request.

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

## Acknowledgements

We acknowledge Kouhei Ohnishi for his kind gift of polyclonal anti-FlgM antibody and Yasuyo Abe and Yoshie Kushima for technical assistance. This work was supported in part by JSPS KAKENHI Grant Numbers JP20K15749 and JP22K06162 (to M.K.), JP19H03182, JP22H02573, and JP22K19274 (to T.M.), and JP21H01772 (to T.U.). This work has also been supported by Platform Project for Supporting Drug Discovery and Life Science Research (BINDS) from AMED under Grant Number JP19am0101117 and JP21am0101117 (to K.N.), by the Cyclic Innovation for Clinical Empowerment (CiCLE) from AMED under Grant Number JP17pc0101020 (to K.N.), and by JEOL YOKOGUSHI Research Alliance Laboratories of Osaka University (to K.N.).

## Author contributions

T.M. and K.N. conceived and designed research; M.K., T.M., and T.U. preformed research; M.K., T.M., and T.U. analysed the data; and T.M. and K.N. wrote the paper based on discussion with other authors.

## Competing interests

The authors declare no competing interests.
