## [Peer Review File · Communications Biology]

Reviewers' comments:

Reviewer #1 (Remarks to the Author):

1. Brief summary of the manuscript

The flagellar type 3 secretion system (FT3S) changes from a selection from rod-hook type protein subunits (RH-type substrates) that make up the rod and hook components of the flagellar hook basal body motor to late-filament type protein substrates (F-type substrates). This process enables to selectively secrete and assemble the RH-type substrates and terminate the hook assembly before switching to a selective mode of secretion of F-type substrates permitting filament assembly. The details of the switching mechanisms are to this date not well understood. There are three main flagellar proteins that are known to be involved in this process: FliK, FlhB and FlhA. This study focusses on the characterization of previously isolated mutants in the highly conserved motif Y369R370L371I272 of FlhA for the ordered protein export by the FT3S and investigate the role of FliJ, FliI and FliH on the export switching function of FlhA.

2. Overall impression of the work

This is a very important question for the field. The T3S system is the only system in Biology that undergoes secretion specificity switch. FlhA, which acts as a secretion chaperone for late substrates, is a central actor in this mechanism. The paper is well written. There is an impressive amount of work with a massive amount of data. The experiments are well designed and the analysis very thorough and complete. The results are of general interest and of great importance to the field.

One thing that made me uncomfortable is the fact that the study started with a * allele with multiple (4) substitutions. Could the authors have defined the most impactful residues in the highly conserved motif Y369R370L371I272 of FlhA instead of mutating all 4 residues at once?*

The FlgL behavior is interesting- do other late secretion substrates have the same gate recognition motif as FlgL that could be tested ?

3. Specific comments, with recommendations for addressing each comment

Abstract: well written.

*Introduction: good overview, necessary to understand these complex machinery
-p4 line 3 and 6 : "specially"- do the authors mean "specifically"?*

- p6- lane 7-13 : I find the flhB (P28T) mutation here distracting because flhB (P28T) could have the effect described in lines 10-13; either talk about the effect of flhB (P28T) alone or omit ?

Results

1) Effect of the AAAA and GGGG mutations on flagellar protein export (p7-8)

Results and main conclusions

The mutations A369A370A371A272 (AAAA) or G369G370G371G272 (GGGG) in the highly conserved motif G368Y369R370L371I272 of FlhA increased secretion of RH-type flagellar substrates but drastically decreased the secretion of F-type flagellar proteins in the AAAA mutant and completely abolished the secretion of F-type flagellar proteins in the GGGG mutant. The 2 mutants formed polyhooks indicating that the AAAA or GGGG mutants cannot complete hook assembly in a timely manner.

Comments

*- p7- line 1-4/ Fig 2a: is it ok to show the motility results since they have already been published?
- p8- line 1-4: "Therefore we conclude that the substrate specificity of FT3SS is determined by the*

conformational state of the FlhAC ring structure..”

Didn't we know that already from previous studies? If so change "conclude" to "confirm".

2) Isolation of up-motile mutants from the AAAA and GGGG mutants (p8)

Results

Motile revertants were isolated from the AAAA and GGGG mutants giving 3 intragenic FlhA mutants AAAT/V; GGGV and 2 extragenic mutants both in FliK (Q338R and A405V)

Comments

Nice!

3) Characterization of intragenic AAAT, AAAT and GGGV suppressor mutants (p9-10)

Results and main conclusions

The FlhA AAAT/V and GGGV mutants shortened the hook lengths compared to the parent AAAA and GGGG strains, but they did not recover the precise hook length control. A fliK deletion inhibited the motility of these mutants, indicating that the FT3SS with the AAAT/V and GGGV mutations are FliK dependant.

-the suppressor mutations AAAT/V and GGGV reduced the secretion of the RH-type flagellar substrates and increased the secretion of the F-type flagellar substrates compared to the AAAA and GGGG parent strains.

- AAAT/V produced longer flagella than the parent AAAA and GGGV produced some short flagellar whereas the GGGG parent did not produced any.

The better export of F-type substrates in the suppressor mutants could be explained by the fact that the FlhA AAAT/V and GGGV mutants can make a more appropriate chaperone binding site in the well conserved dimple of FlhAC once HHB is complete, compared to the AAAA and GGGG parent strains.

Comments

Agreed !

4) Characterization of extragenic suppressor mutants isolated from the AAAA mutant (p11-12)

Results and main conclusions

The FliK suppressors (Q338R and A405V) obtained in the flhA AAAA background resulted in shorter hooks than flhA AAAA with the wild type fliK, indicating that the FT3SS with the AAAA mutation can receive the hook signal more efficiently from a FliK with Q338R or A405V mutations than from the wild type FliK.

-Consistently, the AAAA fliK mutations reduced the secretion of RH-type substrates but did not affect the secretion of F-type substrates compared to the AAAA mutant. They also produced short filaments similar to the AAA parent.

=> altogether these results suggest that the termination of the RH-type protein export and activation of F-type protein export are independent processes: The FlhAC with the AAAA mutation in the ring may efficiently shift its conformation from the RH state to the F-state to terminate rod-hook assembly through the action of the fliK suppressors but can still not produce an appropriate chaperone binding site for the F-type protein export

Comments

- how do Q338R and A405V mutations affect FliK levels?

- do the FliK suppressors (Q338R and A405V) alone give shorter hook with a wild type flhA?

5) Effect of mutations in the GYXLI motif on FlhAC ring formation (p12-13)

Results and main conclusions

-FlhAC-AAAA formed rings whereas FlhAC-GGGG did not. UV-CD measurements revealed that the AAAA and GGGG do not affect secondary structure of FlhA. => a conformational change of the GYXLI motif thus induces a structural transition of the FlhAC ring from a relatively unstable state ring to a more stable F-state ring.

- Intragenic suppressor mutations FlhAC-GGGV formed ring structures whereas FlhAC-GGGV did not. Surprising that FlhAC-GGGV did not since FlhAC-GGGV can secrete F-type substrates. It is possible that the FlhAC-GGGG and FlhAC-GGGV lock FlhAC in the RH state but FlhAC-GGGV can make a structural transition to the F-state when it receives the hook signal from FliK.

Comments

Nice assay!

-p12 line 21: could you define HS-AFM here? you probably define it somewhere else but I don't remember

- p13 line 12: AAAA and GGGG mutations

- p13 line 13: is there a way to test the conformational change of the FlhA GYXLI motif?

-p14 lines 1-2: "So, we hypothesized that the GGGG and GGGV mutations both lock FlhAC in the RH state but FlhAC with the GGGV mutation can make a structural transition to the F state may be with a support of other proteins when it receives the hook length signal from FliK"

I do not understand this sentence- please clarify

6) Effect of mutations in the GYXLI motif on the interaction of FlhAC with the FlgN-FlgK chaperone-substrate complex (p14-15)

Results and main conclusions

-Only a very small amount of FlhACAAAA co-purified with the GST-FlgN/FlgK complex but the FlhACAAAV/T increased the binding affinity for the FlgN/FlgK complex (although not up to the WT level) => since neither FlhACAAAA nor FlhACAAAV/T inhibited ring formation, it is concluded that a proper conformational change of the GYXLI FlhA motif is also required for the formation of an appropriate chaperone binding site in the hydrophobic dimple of FlhAC after FlhAL-C dissociates for the hydrophobic dimple and binds to the D1 and D2 domains of the closest FlhAC subunit in the FlhAC ring.

-FlhAC with the GGGG or GGGV mutations did not co-purified with the GST-FlgN/FlgK complex at all, it is therefore concluded that the GGGG and GGGV mutations stabilize the FlhAC conformation in the RH state, thereby inhibiting the interaction of FlhAC with the FlgN/FlgK in vitro.

Comments

-nice data

7) Effect of mutations in the GYXLI motif on the interaction of FlhAC with FliJ (p15)

Results and main conclusions

In a previous study, it was shown that FlhAL-C W354A and E351A/D356A also inhibited FlhAC ring formation and reduced the binding affinity of FlhAC for flagellar export chaperones in complex with their cognate F-type substrates. These mutants also reduced the binding affinity of FlhAC for FliJ. This lead to the hypothesis that FlhAC-FliJ interaction may be required for efficient transition of the FlhAC-ring structure for the RH state to the F state.

So, in this study, it is shown that GST-FliJ co-purifies with FlhAC AAAA, AAV, GGGG or GGGV mutants the same way as it does with an FlhA wild type, indicating that the flhA mutants do not affect the binding affinity of FlhAC for FliJ. => concluded that the GYXLI motif of FlhA is not involved with the interaction with FliJ.

Comments

-agreed; beautiful gels

8) Effect of FliH and FliI deletions on the export switching function of FlhA with the AAV, AAAT or GGGV mutations

Results and main conclusions

- reduced binding affinity of FlhAC-AAAV and FlhAC-GGGV to FlgN-FlgK chaperone complex
- significant decrease of motility in the AAV, AAAT and GGGV mutants in the absence of FliH and FliI
- decrease of F-type substrate secretion in the double Δ fliHI and FlhA AAV, AAAT and GGGV mutants

compared to $\Delta fliHI$ alone.

- Longer polyhooks observed in the AAV, AAAT and GGGV mutants in the absence of FliH and FliI
- Increase of RH-type substrate secretion in the AAV, AAAT and GGGV mutants in the absence of FliH and FliI compared to $\Delta fliHI$ mutant only.
=> Suggest that the FlhAC ring with the AAV, AAAT, GGGV mutation requires the support of FliH and FliI to efficiently dissociate FlhAL-C from the hydrophobic dimple of FlhAC when it receives the hook length signal for FliK.

Comments

-p16 line 1 and 2: Title : add "s" to FliH and FliI deletions as well as "mutations"

-p16- lines 13-15: "Unlike in the presence of FliH and FliI with or without the B* mutation, the motility of the $\Delta HI-B^*$ AAV, $\Delta HI-B^*$ AAAT, and $\Delta HI-B^*$ GGGV mutants was much worse than that of the $\Delta HI-B^*$ mutant (Fig. 2a, right panel)."

Do you mean that: "The motility of the $\Delta HI-B^*$ mutant is greatly reduced with the presence of AAV, AAAT and GGGV FlhA mutations." ?

-p16- line 19 "Interestingly, much longer polyhooks were frequently observed in these three mutants compared to the AAV, AAAT, and GGGV mutants, respectively (Fig. 2c, lower panel)."

What does much longer polyhook means ? how much longer ?

Do you mean: "The absence of FliH and FliI resulted in significantly longer polyhooks in the FlhA AAV, AAAT or GGGV mutation backgrounds?"

I would suggest to omit the above sentence, and just start with line 19 " we quantitatively measured the polyhook length..."

9) Effect of AAAA and GGGG mutations on FlgL secretion in the presence and absence of FliH and FliI (p17- 19)

Results and main conclusions

-observation of a peculiar change in the level of secretion of FlgL depending on FliHI presence:

- with FliHI, there is less FlgL secretion from the AAAA and GGGG mutants than from the WT or intragenic suppressors (as was the case of the other F-type proteins)

-in the absence of FliHI, FlgL is secreted in larger amount than in the WT or in the suppressor mutants, thus behaving more like a RH-type protein.

=> suggests that the FT3SS with the AAAA or GGGG mutations recognizes FlgL as an RH-type substrate rather than a F-type substrate in the absence of FliI or FliH, while it recognises FlgL as a F-type in the presence of FliHI.

Neither FlgD or FlgL is secreted from the $\Delta fliHI$ cells containing the DflgN allele, but the AAAA and GGGG mutations allowed both FlgD and FlgL to be secreted in the DflgN $\Delta fliHI$ cells

= > means that the AAAA or GGGG mutations do not require FlgN for FlgL secretion in the absence of FlgH and FlgI. The AAAA or GGGG mutations may cause a conformational change in the conserved dimple of FlhAC, allowing the transmembrane export gate complex to become an active protein transporter.

Comments

This is very interesting

-p18 line 2-4: I see that AAAA and GGG mutations impair substrate specificity switching but I do not see experiments about the "timing of hook assembly" in Supplemental Fig 1) . Remove "at appropriate timing of hook assembly" line 3-4 or explain better what is meant

-p18-line 19: I suggest putting the results of Supplementary Figure 8 in the main text as it is an important result for this paper.

-p19-lines 4-8: "Since the *flhA*(D456V) and *flhA*(T490M) mutations in the conserved hydrophobic dimple of FlhAC have been shown to be able to bypass the FlgN defect to a significant degree^{10,28}, the AAAA and GGGG mutations seem to induce a required conformational change in the conserved dimple of FlhAC, allowing the transmembrane export gate complex to become an active protein transporter"

I kind of understand what you mean but if you can clarify the idea, that would be great.

Discussion (p19- 24)

p20- line 12-15: "Because the *flhA*(G368C) mutation in the GYXLI motif has been shown to affect chaperone binding to the conserved hydrophobic dimple of FlhAC^{33,34}, we analyzed the export switching ability of the FlhAC mutants with AAAA and GGGG mutations in the GYXLI motif."

It is not clear to me how this is logical -please rephrase or explain

p22-line 18: replace "much" by "significantly"

Methods

well developed

p29-line 7 : change "Them" by "Then"

Figures

p40- Figure 2

I think it would help the reader to include the results from Supplemental Figure 3 here

p44- Figure 5

what does CBB mean ?

Supplement

p11- Supplementary Table1

missing references 34, 35, 36, 39 and 41

Reviewer #2 (Remarks to the Author):

In this work, Kinoshita et al have examined how a conserved GYXLI motif in the C-terminal region of FlhA influences the ability of 'the conserved dimple' that interacts with chaperone-secretion substrate complexes to switch from secreting rod/hook substrates to filament substrates. A previous study had found that replacement of the GYXLI motif with AAAAA or GGGGG disrupted motility and flagellar biogenesis. Furthermore, previous analysis of mutant with a cysteine in the G position of this motif (flhAG368C) only produces flagella at 30 C and in combination with a fliHI mutant does not produce filaments and instead produces long hooks, suggesting that there may be a link between FliH and FliI and the GYXLI motif to cause a change in FlhA to switch from secretion of rod and hook proteins to filament proteins. In this report, the authors further characterize the FlhA AAAAA and GGGGG mutants for how they impact flagellar biogenesis. These mutants produced long hooks and were less able to transition from secreting rod and hook proteins to filament proteins. Suppressor mutants were identified in this motif and fliK that helped recover the ability of FlhA to transition to secretion of

filament proteins. Analysis of mutants lacking FliH and FliI revealed an absolute dependence on these proteins to enable the GYXLI mutants to switch secretion substrates. A surprising finding was that the FlgL filament substrate was recognized as a rod/hook substrate in the GYXLI mutants lacking FliI and FliH. Thus, this uncovered an ability of the FliI ATPase and FliH spoke structure to correct substrate recognition in these mutants.

This work definitely gets into the 'nuts and bolts' of how certain domains of FlhA with other flagellar proteins function together to cause a switch in secretion of substrates. This work does provide new information for the flagellar biogenesis field. The execution of the work is high quality and the analysis of the mutants with the secretion of substrates and microscopy are very well done. My comments below are mainly to help in some of the presentation of the work to make it easier for readers to access and compare the data presented.

1. The lower part of Figure 1 that shows the RH state and F state of FlhA and the arrangement of the D domains in relation to the linker needs better description in the legend for Figure 1. The main text does not do a sufficient job in describing these changes that occur in this domain of FlhA. Also the drawing of the RH and F states as blocks with the linker moving does not inform how this changes the ability of this domain to recognize different substrates.

2. I think the elements of some figures could be rearranged so that the reader can more easily make comparisons between strains. I don't think the AAAA and GGGG mutants have been examined this thoroughly before, just their motility phenotypes. So, if this is the first time the hook phenotypes have been reported and they need to be compared to all the complemented strains in Figure 2, they should be moved to from figure S1c to Figure 2. In the end, it makes it easier to compare mutants so the reader does not have to flip between Figure 2 and Figure S1c.

3. Can the authors provide any thoughts on what might be special about FlgL compared to FlgK in being recognized as a RH substrate in the GYXLI mutants without FliH and FliI compared to FlgK. Since it can be secreted in the absence of their common chaperone, FlgN, it seems to be something specific about the protein sequence of FlgL. Might there be an evolutionary reason for this ability? I know it will be a manner of speculation, but any thoughts added to the text may make for interesting discussion.

Reviewer #3 (Remarks to the Author):

This mutational analysis of the "GYXLI" motif of FlhA casts additional light on the importance of this segment for the conformation, and possibly conformational changes, of FlhA, with particular reference to the substrate-specificity switch that occurs during flagellar export. The paper's conclusion that this motif is important for the switch, and that the FliH and FliI proteins are also important, is generally well supported, though I don't feel that certain of the very specific, highly detailed conclusions stated in the paper follow uniquely from the data presented. (These concerns are detailed below). The paper provides new and useful data on FlhA and its interplay with other components of the apparatus.

Some specific questions and concerns:

minamino strictly ordered

1. Given that function of the AAAA and GGGG mutants can be partially rescued by mutations in just the 4th position, it's natural to wonder if mutation of that fourth position suffices to give the phenotype. So I was surprised to see that single-residue replacements (with A or G) in that evidently critical position have not been characterized.

2. In the Introduction and a number of other places, the notion of proofreading is introduced, it seems to me rather abruptly. I'm not sure whether it's an applicable term here, even if FliHI assist FlhA (and other components) in bringing about a reliable switch in specificity. Is there any evidence that when an "error" occurs, the system actively reverses that action? (which is what proofreading would do)

3. There are a few places where a rather detailed and specific conclusion is drawn, when I feel that the data support only more general surmises. Examples are:

p. 8 line 1. This conclusion seems to go beyond what the evidence supports at this point. A fairly substantial disruption of FlhA (the quadruple mutants) compromises the switch from early to late substrate specificity. Does it follow that the specificity is determined by the conformational state of the FlhAc ring? As phrased, the suggestion is being made that it is this, primarily, that determines the specificity. But what if the FlhA ring supports a needed conformation of FlhB, which in turn dictates specificity? Or is needed for proper docking of FliI, which in turn influences specificity? Similar concerns with statement beginning line 4. This I might rephrase to say that the GLRYI motif is an important determinant of FlhAc conformation, based on earlier work, so it is not surprising that mutating it will alter FlhAc function. (But, I would add, in a way that is not entirely clear at this point.)

page 10 line 16 similar leap. The switch to late has been largely rescued; it doesn't necessarily follow that the reason for this is that a more appropriate chaperone binding site has been restored. Maybe so, but perhaps it involves something else, or something in addition. Line 19 extends this line of thinking, again without what I would call clear justification. The ring moves from the RH state to the F state, but I'm not sure what data here says that it also, as a distinct phenomenon, forms an appropriate chaperone binding site. Having the site is a characteristic of the F state. This presentation makes it sound as if the site itself has been directly implicated, but I don't think this is the case (though its involvement is likely).

p. 13 line 13 similarly. the mutations affect stability of the ring. Does it follow that the key feature of the RH state is ring instability?

p. 15 line 6 again. the mutations prevent the binding. Does it follow that they lock it into the RH state? Other states might exist, particularly in a heavily mutated, possibly aberrantly flexible, protein. It seems more likely that the mutations destabilize a needed state, in this case the state with a well-formed hydrophobic dimple.

p. 15 line 14, also p. 17 line 9, again what I think might be overly specific interpretations. Loss of HI impairs. The AAAV etc mutants are somewhat impaired. Combining the two leads to a more severe defect.

p. 17 line 9 another example: highly specific, detailed interpretation not uniquely supported by the data.

p. 19 line 4 another very specific conclusion that I'm not entirely convinced of. As a conservative default position, it might be reasonable to suppose that the quadruple mutants will make the protein structure less defined, possibly more malleable. To conclude from the phenotypes that the mutations induce a required conformational change seems a much less conservative interpretation.

4. p. 12 line 5. I found this somewhat confusing; if it is switched "to the F state...", doesn't that mean that the chaperone binding site is formed?

5. p. 13 line 11. I worry that CD measurements lack the resolution to rule out the possibility that the

mutations have affected the conformation of the protein at the inter-subunit interface. The motif is buried right under a part of the protein that participates directly in interaction with the neighboring subunit; it would be surprising if a quadruple mutant, affecting some largely buried side-chains, did not affect conformation in this region. The relevant regions are not likely to be large contributors to the CD signals of the protein; i.e. significant stretches of alpha-helix don't appear to be involved. So I think it does not follow that the GLXYI motif must itself undergo a conformational change that initiates subsequent events. It is true that its conformation, and likely its conformational plasticity, are probably important. But it's going farther to say that it "induces" conformational responses.

6. Sup fig 8; ability to export FlgL without FlgN, in the GGGG or AAAA mutants, is notable. Is this also true for FlgK? Related: The export of FlgK responds to loss of HI (and the B* mutation) very differently than FlgL. Comments or thoughts on this difference?

7. p. 16 line 19. It is true that B* by itself doesn't change export much. It does not follow that the B* mutation does not play an important role in the changes that are seen when HI are deleted, though. If involvement of B* is to be ruled out to justify the present focus on the role of HI, then the HI deletion needs to be made in the presence of wild-type FlhB. I think the results will be quite different, because the B* mutation (I think) is necessary for export to proceed much at all in the absence of HI. If I recall correctly the B* mutation was isolated as a suppressor of the HI deletion, and it seems to make export more permissive. The strengthened phenotypes seen in the HI-B* background are in this sense not surprising, and could be due as much to B* as to loss of HI.

8. p. 17 line 17. Can some comment be made about the FlgK row? FlgK export is practically prevented by the delHI-B* mutations, and contrasts very strongly with FlgL.

9. p. 17, "with or without B*" The statement is true, but pertains to the case with HI present; probably more relevant is whether B* makes any difference when HI is deleted. Likely it does, and in that case, we can't separate the del-HI effect from the B* effect (because B* is always introduced along with delHI in the experiments here).

10. p. 20 line 1. Again I wondered why the term 'proofreading' is used.

11. p. 22, the interpretation of the fliK extragenic suppressors. Alternatively, FlhAc might assist in the FliK-induced dissociation of a part of FlhBc (something that is thought to occur during the specificity switch!), and the AAAA and GGGG mutations impede this. In this alternative interpretation, the FliK* mutations strengthen the ability of FliK to induce the dissociation of the MotBc bit by itself, for example by slowing the export of FliK so that it has more time to induce the dissociation. This raises another: why is the FlhBcc (C-terminal part of the FlhB C-terminal domain) dissociation model not discussed here?

12. bottom p. 23. The mutation is suggested to restrict a specific process (domain rotation), whereas it seems more likely (to me) that it creates a more permissive, conformationally fluid situation, where a particular, needed conformation is less stabilized than in the wild type.

Our responses are listed below. We highlighted all changes in the revised manuscript (Marked Up version).

To Reviewer #1

1. Brief summary of the manuscript

The flagellar type 3 secretion system (FT3S) changes from a selection from rod-hook type protein subunits (RH-type substrates) that make up the rod and hook components of the flagellar hook basal body motor to late-filament type protein substrates (F-type substrates). This process enables to selectively secrete and assemble the RH-type substrates and terminate the hook assembly before switching to a selective mode of secretion of F-type substrates permitting filament assembly. The details of the switching mechanisms are to this date not well understood. There are three main flagellar proteins that are known to be involved in this process: FliK, FlhB and FlhA. This study focusses on the characterization of previously isolated mutants in the highly conserved motif Y369R370L371I272 of FlhA for the ordered protein export by the FT3S and investigate the role of FliJ, FliI and FliH on the export switching function of FlhA.

Re: Thank you so much for your deep understanding of our research.

2. Overall impression of the work

This is a very important question for the field. The T3S system is the only system in Biology that undergoes secretion specificity switch. FlhA, which acts as a secretion chaperone for late substrates, is a central actor in this mechanism. The paper is well written. There is an impressive amount of work with a massive amount of data. The experiments are well designed and the analysis very thorough and complete. The results are of general interest and of great importance to the field.

Re: Thank you so much for your supportive comments.

One thing that made me uncomfortable is the fact that the study started with a  allele with multiple (4) substitutions. Could the authors have defined the most impactful residues in the highly conserved motif Y369R370L371I372 of FlhA instead of mutating all 4 residues at once?

Re: The AAV/T and GGGV mutants were isolated as up-motile mutants from the AAAA and GGGG mutant, respectively, suggesting that the hydrophobic side chain of Ile-372 is very important for the export switch function of FlhA. To confirm this, we constructed the *flhA(I372A)* and *flhA(I372G)* mutants (**See Figure 5 in the revised manuscript**). The *flhA(I372A)* mutation reduced the motility in soft agar whereas the *flhA(I372G)* mutation inhibited the motility. The *flhA(I372G)* mutation inhibited substrate specificity switching of the FT3SS, thereby increasing the secretion levels of

RH-type substrates such as FlgD and FliK and reducing the secretion levels of F-type substrates such as FlgK, FlgL, and FliC. In the absence of FliH and FliI, these two mutations significantly inhibited the substrate specificity switch in a manner similar to the AAV/T and GGGV mutations. Therefore, Ile-372 is the most impactful residue for efficient and robust structural remodeling of the FlhA_C ring responsible for export switching of the FT3SS.

The FlgL behavior is interesting- do other late secretion substrates have the same gate recognition motif as FlgL that could be tested ?

Re: It has been reported that the export signal of FliC recognized by the *Salmonella* FT3SS is located within residues 26–47 in FliC (Végh, B. M., Gál, P., Dobó, J., Závodszky, P., and Vonderviszt, F. Localization of the flagellum-specific secretion signal in *Salmonella* flagellin. *Biochem Biophys Res Commun* 345: 93-98, 2006). Therefore, we investigated whether the F-type substrates have a common motif in a way similar to the RH-type substrates. However, we found no common motif in the N-terminal region of the F-type substrates.

3. Specific comments, with recommendations for addressing each comment

Abstract: well written.

Re: Thank you so much. The number of words in the abstract in our original manuscript exceeded 150 words, and so we have shortened the abstract in the revised manuscript. However, we believe that it is sufficient for the readers to understand our important discovery.

Introduction: good overview, necessary to understand these complex machinery

Re: Thank you so much. We have shortened the Introduction in the revised manuscript, because the number of words in the introduction, Results, and Discussion in our original manuscript exceeded 5,000 words. However, we believe that it is sufficient for non-expert readers to understand the general background of our research.

-p4 line 3 and 6 : “specially”- do the authors mean “specifically”?

Re: We corrected.

- p6- lane 7-13 : I find the flhB (P28T) mutation here distracting because flhB (P28T) could have the effect described in lines 10-13; either talk about the effect of flhB (P28T) alone or omit ?

Re: We described the effect of the *flhB*(P28T) mutation alone as follows:

“The *flhB*(P28T) mutation alone does not affect the hook length control at all³⁶”

Results

1) Effect of the AAAA and GGGG mutations on flagellar protein export (p7-8)

Results and main conclusions

The mutations A369A370A371A272 (AAAA) or G369G370G371G272 (GGGG) in the highly conserved motif G368Y369R370L371I272 of FlhA increased secretion of RH-type flagellar substrates but drastically decreased the secretion of F-type flagellar proteins in the AAAA mutant and completely abolished the secretion of F-type flagellar proteins in the GGGG mutant. The 2 mutants formed polyhooks indicating that the AAAA or GGGG mutants cannot complete hook assembly in a timely manner.

Comments

- p7- line 1-4/ Fig 2a: is it ok to show the motility results since they have already been published?

Re: For the benefit of readers, the motility data of the AAAA and GGGG mutants are necessary as controls for up-motile mutants isolated from these two mutants.

- p8- line 1-4: “Therefore we conclude that the substrate specificity of FT3SS is determined by the conformational state of the FlhAC ring structure.”
Didn't we know that already from previous studies? If so change “conclude” to “confirm”.

Re: No, we did not know it. This is the first time that the *flhA* mutants have been shown to produce polyhooks like the *fliK* null mutant. Nonetheless, we changed “conclude” to “suggest”.

2) Isolation of up-motile mutants from the AAAA and GGGG mutants (p8)

Results

Motile revertants were isolated from the AAAA and GGGG mutants giving 3 intragenic FlhA mutants AAAT/V; GGGV and 2 extragenic mutants both in FliK (Q338R and A405V)

Comments

Nice!

Re: Thank you.

3) Characterization of intragenic AAATV, AAAT and GGGV suppressor mutants (p9-10)

Results and main conclusions

The FlhA AAAT/V and GGGV mutants shortened the hook lengths compared to the

parent AAAA and GGGG strains, but they did not recover the precise hook length control. A *fliK* deletion inhibited the motility of these mutants, indicating that the *fT3SS* with the AAAT/V and GGGV mutations are *FliK* dependant.

-the suppressor mutations AAAT/V and GGGV reduced the secretion of the RH-type flagellar substrates and increased the secretion of the F-type flagellar substrates compared to the AAAA and GGGG parent strains.

- AAAT/V produced longer flagella than the parent AAAA and GGGV produced some short flagellar whereas the GGGG parent did not produced any.

The better export of F-type substrates in the suppressor mutants could be explained by the fact that the *FlhA* AAAT/V and GGGV mutants can make a more appropriate chaperone binding site in the well conserved dimple of *FlhAC* once HHB is complete, compared to the AAAA and GGGG parent strains.

Comments

Agreed !

Re: Thank you.

4) Characterization of extragenic suppressor mutants isolated from the AAAA mutant (p11-12)

Results and main conclusions

The *FliK* suppressors (Q338R and A405V) obtained in the *flhA* AAAA background resulted in shorter hooks than *flhA* AAAA with the wild type *fliK*, indicating that the *fT3SS* with the AAAA mutation can receive the hook signal more efficiently from a *FliK* with Q338R or A405V mutations than from the wild type *FliK*.

-Consistently, the AAAA *fliK* mutations reduced the secretion of RH-type substrates but did not affect the secretion of F-type substrates compared to the AAAA mutant. They also produced short filaments similar to the AAA parent.

=> altogether these results suggest that the termination of the RH-type protein export and activation of F-type protein export are independent processes: The *FlhAC* with the AAAA mutation in the ring may efficiently shift its conformation from the RH state to the F-state to terminate rod-hook assembly through the action of the *fliK* suppressors but can still not produce an appropriate chaperone binding site for the F-type protein export

Comments

- how do Q338R and A405V mutations affect *FliK* levels?

Re: These two mutations did not affect the cellular and extracellular levels of *FliK* at all (See Fig. 4d, 3rd row in the revised manuscript).

- do the *FliK* suppressors (Q338R and A405V) alone give shorter hook with a wild type *flhA*?

Re: These *fliK* mutations alone did not affect the hook length control (**See Supplementary Fig. 4c in the revised manuscript**)

5) Effect of mutations in the GYXLI motif on FlhAC ring formation (p12-13)

Results and main conclusions

-FlhAC-AAAA formed rings whereas FlhAC-GGGG did not. UV-CD measurements revealed that the AAAA and GGGG do not affect secondary structure of FlhA. => a conformational change of the GYXLI motif thus induces a structural transition of the FlhAC ring from a relatively unstable state ring to a more stable F-state ring.

- Intragenic suppressor mutations FlhAC-GGGV formed ring structures whereas FlhAC-GGGV did not. Surprising that FlhAC-GGGV did not since FlhAC-GGGV can secrete F-type substrates. It is possible that the FlhAC-GGGG and FlhAC-GGGV lock FlhAC in the RH state but FlhAC-GGGV can make a structural transition to the F-state when it receives the hook signal from *FliK*.

Comments

Nice assay!

Re: Thank you.

-p12 line 21: could you define HS-AFM here? you probably define it somewhere else but I don't remember

Re: The definition of HS-AFM was given in the Introduction of our original manuscript. But to help readers, we gave it in the Result section.

- p13 line 12: AAAA and GGGG mutations

Re: Corrected

- p13 line 13: is there a way to test the conformational change of the FlhA GYXLI motif?

Re: We previously showed that the conserved GYXLI motif in FlhA_C is critical for cyclic open-close domain motions of FlhA_C that is important for FlhA function and that the conformational flexibility of the GYXLI motif is important for such dynamic domain motions (**Inoue et al. Structure 2019; Minamino et al. Microbiol. Spectr. 2022**). To confirm this, we compared the conformation of this motif in different crystal structures of FlhA_C and found that it actually takes different conformations (**See Supplementary**

Fig. 2 in the revised manuscript). Consistently, the binding of FlhT induces a conformational change of the conserved GYXLI motif of FlhA_C (**See Supplementary Fig. 7c in the revised manuscript**). Therefore, we suggest that an appropriate conformational change in the GYXLI motif is required for FlhA_{L-C} to efficiently bind to the D1 and D3 domains of the closest subunit in the ring.

-p14 lines1-2: "So, we hypothesized that the GGGG and GGGV mutations both lock FlhAC in the RH state but FlhAC with the GGGV mutation can make a structural transition to the F state may be with a support of other proteins when it receives the hook length signal from FliK"

I do not understand this sentence- please clarify

Re: We changed this sentence as follows:

"So, we hypothesized that the GGGG and GGGV mutations both lock FlhA_C in the RH state but FlhA_C with the GGGV mutation may be able to make the transition from the RH state to the F state with a support of other proteins when it receives the hook length signal from FliK."

6) Effect of mutations in the GYXLI motif on the interaction of FlhAC with the FlgN-FlgK chaperone-substrate complex (p14-15)

Results and main conclusions

-Only a very small amount of FlhACAAAA co-purified with the GST-FlgN/FlgK complex but the FlhACAAAV/T increased the binding affinity for the FlgN/FlgK complex (although not up to the WT level) => since neither FlhACAAAA nor FlhACAAAV/T inhibited ring formation, it is concluded that a proper conformational change of the GYXLI FlhA motif is also required for the formation of an appropriate chaperone binding site in the hydrophobic dimple of FlhAC after FlhAL-C dissociates for the hydrophobic dimple and binds to the D1 and D2 domains of the closest FlhAC subunit in the FlhAC ring.

-FlhAC with the GGGG or GGGV mutations did not co-purified with the GST-FlgN/FlgK complex at all, it is therefore concluded that the GGGG and GGGV mutations stabilize the FlhAC conformation in the RH state, thereby inhibiting the interaction of FlhAC with the FlgN/FlgK in vitro.

Comments

-nice data

Re: Thank you.

7) Effect of mutations in the GYXLI motif on the interaction of FlhAC with FliJ (p15)

Results and main conclusions

In a previous study, it was shown that FlhAL-C W354A and E351A/D356A also inhibited FlhAC ring formation and reduced the binding affinity of FlhAC for flagellar export chaperones in complex with their cognate F-type substrates. These mutants also reduced the binding affinity of FlhAC for FliJ. This lead to the hypothesis that FlhAC-FliJ interaction may be required for efficient transition of the FlhAC-ring structure for the RH state to the F state.

So, in this study, it is shown that GST-FliJ co-purifies with FlhAC AAAA, AAV, GGGG or GGGV mutants the same way as it does with an FlhA wild type, indicating that the flhA mutants do not affect the binding affinity of FlhAC for FliJ. => concluded that the GYXLI motif of FlhA is not involved with the interaction with FliJ.

Comments

-agreed; beautiful gels

Re: Thank you.

8) Effect of FliH and FliI deletions on the export switching function of FlhA with the AAV, AAAT or GGGV mutations

Results and main conclusions

- reduced binding affinity of FlhAC-AAV and FlhAC-GGGV to FlgN-FlgK chaperone complex*
 - significant decrease of motility in the AAV, AAAT and GGGV mutants in the absence of FliH and FliI*
 - decrease of F-type substrate secretion in the double $\Delta fliHI$ and FlhA AAV, AAAT and GGGV mutants compared to $\Delta fliHI$ alone.*
 - Longer polyhooks observed in the AAV, AAAT and GGGV mutants in the absence of FliH and FliI*
 - Increase of RH-type substrate secretion in the AAV, AAAT and GGGV mutants in the absence of FliH and FliI compared to $\Delta fliHI$ mutant only.*
- =>Suggest that the FlhAC ring with the AAV, AAAT, GGGV mutation requires the support of FliH and FliI to efficiently dissociate FlhAL-C from the hydrophobic dimple of FlhAC when it receives the hook length signal for FliK.*

Comments

-p16 line 1 and 2: Title : add "s" to FliH and FliI deletions as well as "mutations"

Re: We change it to "Removal of both FliH and FliI"

-p16- lines 13-15: "Unlike in the presence of FliH and FliI with or without the B mutation, the motility of the $\Delta HI-B^*$ AAV, $\Delta HI-B^*$ AAAT, and $\Delta HI-B^*$ GGGV mutants was much worse than that of the $\Delta HI-B^*$ mutant (Fig. 2a, right panel)."*

Do you mean that: "The motility of the $\Delta HI-B^$ mutant is greatly reduced with the*

presence of AAV, AAAT and GGGV FlhA mutations.” ?

Re: No, we do not. To avoid confusion, we changed this sentence as follows:

“Unlike in the presence of FliH and FliI, the motility of the Δ HI-B AAV, Δ HI-B* AAAT, and Δ HI-B* GGGV mutants was worse than that of the Δ HI-B* mutant (Fig. 3a, right panel).”*

-p16- line 19 “Interestingly, much longer polyhooks were frequently observed in these three mutants compared to the AAV, AAAT, and GGGV mutants, respectively (Fig. 2c, lower panel).”

What does much longer polyhook means ? how much longer ?

Do you mean:” The absence of FliH and FliI resulted in significantly longer polyhooks in the FlhA AAV, AAAT or GGGV mutation backgrounds?”

Re: They were two to three times longer as indicated in the following sentences.
Yes, you understand what we meant correctly.

I would suggest to omit the above sentence, and just start with line 19 “ we quantitatively measured the polyhook length...”

Re: We agreed and so decided to describe only the polyhook lengths of the Δ HI-B* AAV, Δ HI-B* AAAT, and Δ HI-B* GGGV mutants in the revised manuscript.

9) Effect of AAAA and GGGG mutations on FlgL secretion in the presence and absence of FliH and FliI (p17- 19)

Results and main conclusions

-observation of a peculiar change in the level of secretion of FlgL depending on FliHI presence:

- with FliHI, there is less FlgL secretion from the AAAA and GGGG mutants than from the WT or intragenic suppressors (as was the case of the other F-type proteins)

-in the absence of FliHI, FlgL is secreted in larger amount than in the WT or in the suppressor mutants, thus behaving more like a RH-type protein.

=> suggests that the FT3SS with the AAAA or GGGG mutations recognizes FlgL as an RH-type substrate rather than a F-type substrate in the absence of FliI or FliH, while it recognises FlgL as a F-type in the presence of FliHI.

Neither FlgD or FlgL is secreted from the Δ fliHI cells containing the DflgN allele, but the AAAA and GGGG mutations allowed both FlgD and FlgL to be secreted in the DflgN DfliHI cells

= >means that the AAAA or GGGG mutations do not require FlgN for FlgL secretion in the absence of FlgH and FlgI. The AAAA or GGGG mutations may cause a

conformational change in the conserved dimple of FlhAC, allowing the transmembrane export gate complex to become an active protein transporter.

Comments

This is very interesting

Re: Thank you.

-p18 line 2-4: I see that AAAA and GGG mutations impair substrate specificity switching but I do not see experiments about the “timing of hook assembly” in Supplemental Fig 1). Remove “at appropriate timing of hook assembly” line 3-4 or explain better what is meant

Re: To avoid confusion, we changed this sentence as follows.

“Because the AAAA and GGGG mutations inhibit substrate specificity switching of the FT3SS from the RH-type to the F-type even in the presence of FliH and FliI, thereby producing polyhooks (Fig. 2)”

-p18-line 19: I suggest putting the results of Supplementary Figure 8 in the main text as it is an important result for this paper.

Re: Thank you for your suggestion, and we did so.

*-p19-lines 4-8: “Since the flhA(D456V) and flhA(T490M) mutations in the conserved hydrophobic dimple of FlhAC have been shown to be able to bypass the FlgN defect to a significant degree^{10,28}, the AAAA and GGGG mutations seem to induce a required conformational change in the conserved dimple of FlhAC, allowing the transmembrane export gate complex to become an active protein transporter”
I kind of understand what you mean but if you can clarify the idea, that would be great.*

Re: We rephrased this part as follows:

“The flhA(D456V) and flhA(T490M) mutations in the conserved hydrophobic dimple of FlhAC have been shown to be able to bypass the FlgN defect to a significant degree^{10,28}. Because the flhA(G368C) mutation in the GYXLI motif affects a conformation of the hydrophobic dimple^{33,34}, the AAAA and GGGG mutations may induce a required conformational change in the conserved dimple of FlhAC to allow the transmembrane export gate complex to become an active protein transporter.”

Discussion (p19- 24)

p20- line 12-15: “Because the flhA(G368C) mutation in the GYXLI motif has been shown to affect chaperone binding to the conserved hydrophobic dimple of FlhA_C^{33,34}, we analyzed the export switching ability of the FlhA_C mutants with AAAA and GGGG mutations in the GYXLI motif.”

It is not clear to me how this is logical -please rephrase or explain

Re: We rephrased this part as follows for clarity:

“The highly conserved GYXLI motif of FlhA_C acts as a structural switch to facilitate cyclic domain motions of FlhA_C through periodically remodeling its hydrophobic side-chain interaction networks³⁴. The *flhA(G368C)* mutation in the GYXLI motif affects substrate specificity switching of the FT3SS from the RH-type to the F-type in the absence of FliH and FliI. Furthermore, this mutation reduces the binding affinity of FlhA_C for export chaperones in complex with their cognate F-type substrates^{33,34}. These observations raised the possibility that the GYXLI motif is also involved in substrate specificity switching of the FT3SS from the RH-type to the F-type. Therefore, we analyzed the effect of the AAAA and GGGG mutations in the GYXLI motif on the export switching function of the FT3SS.”

p22-line 18: replace “much” by “significantly”

Re: We change it to “nearly twice”.

Methods

well developed

Re: Thank you.

p29-line 7 : change “Them” by “Then”

Re: Corrected.

Figures

p40- Figure 2

I think it would help the reader to include the results from Supplemental Figure 3 here

Re: Thank you for the suggestion, and we did so.

p44- Figure 5

what does CBB mean ?

Re: CBB means “Coomassie Brilliant blue”. We changed “CBB” to “Coomassie Brilliant blue”.

Supplement

p11- Supplementary Table1

missing references 34, 35, 36, 39 and 41

Re: These references are listed in the main text. However, to avoid confusion, this table was moved from the Supplementary Information file to the main text of the revised manuscript.

To Reviewer #2

In this work, Kinoshita et al have examined how a conserved GYXLI motif in the C-terminal region of FlhA influences the ability of ‘the conserved dimple’ that interacts with chaperone-secretion substrate complexes to switch from secreting rod/hook substrates to filament substrates. A previous study had found that replacement of the GYXLI motif with AAAAA or GGGGG disrupted motility and flagellar biogenesis. Furthermore, previous analysis of mutant with a cysteine in the G position of this motif (flhAG368C) only produces flagella at 30 C and in combination with a fliHI mutant does not produce filaments and instead produces long hooks, suggesting that there may be a link between FliH and FliI and the GYXLI motif to cause a change in FlhA to switch from secretion of rod and hook proteins to filament proteins. In this report, the authors further characterize the FlhA AAAAA and GGGGG mutants for how they impact flagellar biogenesis. These mutants produced long hooks and were less able to transition from secreting rod and hook proteins to filament proteins. Suppressor mutants were identified in this motif and fliK that helped recover the ability of FlhA to transition to secretion of filament proteins. Analysis of mutants lacking FliH and FliI revealed an absolute dependence on these proteins to enable the GYXLI mutants to switch secretion substrates. A surprising finding was that the FlgL filament substrate was recognized as a rod/hook substrate in the GYXLI mutants lacking FliI and FliH. Thus, this uncovered an ability of the FliI ATPase and FliH spoke structure to correct substrate recognition in these mutants.

This work definitely gets into the ‘nuts and bolts’ of how certain domains of FlhA with other flagellar proteins function together to cause a switch in secretion of substrates. This work does provide new information for the flagellar biogenesis field. The execution of the work is high quality and the analysis of the mutants with the secretion of

substrates and microscopy are very well done. My comments below are mainly to help in some of the presentation of the work to make it easier for readers to access and compare the data presented.

Re: Thank you so much for your supportive comments.

1. The lower part of Figure 1 that shows the RH state and F state of FlhA and the arrangement of the D domains in relation to the linker needs better description in the legend for Figure 1. The main text does not do a sufficient job in describing these changes that occur in this domain of FlhA. Also the drawing of the RH and F states as blocks with the linker moving does not inform how this changes the ability of this domain to recognize different substrates.

Re: We modified the lower part of Figure 1 and Figure legend as well as the Introduction to allow the readers to more easily understand the export switching mechanism of the FT3SS.

2. I think the elements of some figures could be rearranged so that the reader can more easily make comparisons between strains. I don't think the AAAA and GGGG mutants have been examined this thoroughly before, just their motility phenotypes. So, if this is the first time the hook phenotypes have been reported and they need to be compared to all the complemented strains in Figure 2, they should be moved to from figure S1c to Figure 2. In the end, it makes it easier to compare mutants so the reader does not have to flip between Figure 2 and Figure S1c.

Re: Thank you so much for your comments. We agree that this is the first time that the AAAA and GGGG mutants exhibit the polyhook-filament and polyhook phenotypes, respectively. Because original Figure 2 was already too big to incorporate EM images, we decided to include Figure S1 from the Supplementary Information file as new Figure 2 in the main text of the revised manuscript. We believe that this will allow readers to easily compare the phenotypes of mutants.

3. Can the authors provide any thoughts on what might be special about FlgL compared to FlgK in being recognized as a RH substrate in the GYXLI mutants without FliH and FliI compared to FlgK. Since it can be secreted in the absence of their common chaperone, FlgN, it seems to be something specific about the protein sequence of FlgL. Might there be an evolutionary reason for this ability? I know it will be a manner of speculation, but any thoughts added to the text may make for interesting discussion.

Re: As described in our response to Reviewer #1, we analyzed the N-terminal amino acid sequences of F-type substrates but did not obtain any clue to explain this

interesting phenomenon. So, we have no idea at this moment. Because the guidelines of Communications Biology require that the text including Introduction, Results, and Discussion should be no more than 5,000 words, and we do not want to mislead the readers, we have chosen not to describe our speculation on the difference between FlgK and FlgL.

To Reviewer #3

This mutational analysis of the "GYXLI" motif of FlhA casts additional light on the importance of this segment for the conformation, and possibly conformational changes, of FlhA, with particular reference to the substrate-specificity switch that occurs during flagellar export. The paper's conclusion that this motif is important for the switch, and that the FliH and FliI proteins are also important, is generally well supported, though I don't feel that certain of the very specific, highly detailed conclusions stated in the paper follow uniquely from the data presented. (These concerns are detailed below). The paper provides new and useful data on FlhA and its interplay with other components of the apparatus.

Re: Thank you so much for your supportive comments.

Some specific questions and concerns:

minamino strictly ordered

1. Given that function of the AAAA and GGGG mutants can be partially rescued by mutations in just the 4th position, it's natural to wonder if mutation of that fourth position suffices to give the phenotype. So I was surprised to see that single-residue replacements (with A or G) in that evidently critical position have not been characterized.

Re: As we responded to Reviewer #1, we constructed the *flhA(I372A)* and *flhA(I372G)* mutants and found that the *flhA(I372A)* mutant showed reduced motility in soft agar whereas the *flhA(I372G)* mutant exhibited a non-motile phenotype. The I372G substitution significantly affected substrate specificity switching of the FT3SS from the RH-type to the F-type, thereby producing polyhooks (**See Figure 5 in the revised manuscript**). These results suggest that the hydrophobic side chain of Ile-372 is important for the substrate specificity switch. We included these results in the revised manuscript.

2. In the Introduction and a number of other places, the notion of proofreading is introduced, it seems to me rather abruptly. I'm not sure whether it's an applicable term here, even if FliHI assist FlhA (and other components) in bringing about a reliable

switch in specificity. Is there any evidence that when an "error" occurs, the system actively reverses that action? (which is what proofreading would do)

Re: We removed "proofreading" from the entire text.

3. There are a few places where a rather detailed and specific conclusion is drawn, when I feel that the data support only more general surmises. Examples are:

Re: As this reviewer noted, we decided to avoid the use of "conclude".

p. 8 line 1. This conclusion seems to go beyond what the evidence supports at this point. A fairly substantial disruption of FlhA (the quadruple mutants) compromises the switch from early to late substrate specificity. Does it follow that the specificity is determined by the conformational state of the FlhAc ring? As phrased, the suggestion is being made that it is this, primarily, that determines the specificity. But what if the FlhA ring supports a needed conformation of FlhB, which in turn dictates specificity? Or is needed for proper docking of Flil, which in turn influences specificity? Similar concerns with statement beginning line 4. This I might rephrase to say that the GLRYI motif is an important determinant of FlhAc conformation, based on earlier work, so it is not surprising that mutating it will alter FlhAc function. (But, I would add, in a way that is not entirely clear at this point.)

Re: We changed our statement as follows:

"Because these two mutants produce polyhooks even when both FliK and FlhB are intact, we suggest that the substrate specificity of FT3SS is determined by the conformational state of the FlhAc ring."

page 10 line 16 similar leap. The switch to late has been largely rescued; it doesn't necessarily follow that the reason for this is that a more appropriate chaperone binding site has been restored. Maybe so, but perhaps it involves something else, or something in addition. Line 19 extends this line of thinking, again without what I would call clear justification. The ring moves from the RH state to the F state, but I'm not sure what data here says that it also, as a distinct phenomenon, forms an appropriate chaperone binding site. Having the site is a characteristic of the F state. This presentation makes it sound as if the site itself has been directly implicated, but I don't think this is the case (though its involvement is likely).

Re: We changed our statement as follows:

“The conformational change of the GYXLI motif of FlhA_C would be necessary not only for efficient transition of the FlhA_C ring from the RH state to the F state but also for the formation of appropriate chaperone binding sites in the ring.”

p. 13 line 13 similarly. the mutations affect stability of the ring. Does it follow that the key feature of the RH state is ring instability?

Re: We change our statement as follows:

“we suggest that an appropriate conformational change in the GYXLI motif is required for FlhA_{L-C} to bind to the D1 and D3 domains of the closest subunit in the ring.”

p. 15 line 6 again. the mutations prevent the binding. Does it follow that they lock it into the RH state? Other states might exist, particularly in a heavily mutated, possibly aberrantly flexible, protein. It seems more likely that the mutations Destabilize a needed state, in this case the state with a well-formed hydrophobic dimple.

Re: We changed our statement as follows:

“Therefore, the GGGG and GGGV mutations might stabilize the FlhA_C conformation in the RH state, thereby inhibiting the interaction of FlhA_C with the FlgN-FlgK complex *in vitro*.”

p. 15 line 14, also p. 17 line 9, again what I think might be overly specific interpretations. Loss of HI impairs. The AAVV etc mutants are somewhat impaired. Combining the two leads to a more severe defect.

Re: We changed our statement as follows:

“It has been proposed that an interaction between FlhA_C and FliJ may be required for efficient transition of the FlhA_C ring from the RH state to the F state^{22,31}.”

p. 17 line 9 another example: highly specific, detailed interpretation not uniquely supported by the data.

Re: We also analyzed the effect of removal of both FliH and FliI on the export switching function of the *flhA(I372A)* and *flhA(I372G)* mutants and found that these two mutant variants of FlhA requires FliH and FliI to fully exert their export switching function (**See Figure 5b, right panel**). Therefore, we change our statement as follows:

“Furthermore, the *flhA(I372A)* and *flhA(I372G)* mutations inhibited substrate specificity switching of the FT3SS from the RH-type to the F-type in the absence of FliH and FliI,

thereby inhibiting the motility of the Δ HI-B* mutant (Fig. 5b, right panels). These results suggest that the FlhA_C ring requires the support of FliH and FliI to efficiently undergo a structural transition from the RH state to the F state.”

p. 19 line 4 another very specific conclusion that I'm not entirely convinced of. As a conservative default position, it might be reasonable to suppose that the quadruple mutants will make the protein structure less defined, possibly more malleable. To conclude from the phenotypes that the mutation induce a required conformational change seems a much less conservative interpretation.

Re: We change our statement as follows:

“The *flhA*(D456V) and *flhA*(T490M) mutations in the conserved hydrophobic dimple of FlhA_C have been shown to be able to bypass the FlgN defect to a significant degree^{10,28}. Because the *flhA*(G368C) mutation in the GYXLI motif affects a conformation of the hydrophobic dimple^{33,34}, the AAAA and GGGG mutations may induce a required conformational change in the conserved dimple of FlhA_C to allow the transmembrane export gate complex to become an active protein transporter.”

4. p. 12 line 5. I found this somewhat confusing; if it is switched "to the F state...", doesn't that mean that the chaperone binding site is formed?

Re: To avoid confusion, we deleted this statement from the text.

5. p. 13 line 11. I worry that CD measurements lack the resolution to rule out the possibility that the mutations have affected the conformation of the protein at the inter-subunit interface. The motif is buried right under a part of the protein that participates directly in interaction with the neighboring subunit; it would be surprising if a quadruple mutant, affecting some largely buried side-chains, did not affect conformation in this region. The relevant regions are not likely to be large contributors to the CD signals of the protein; i.e. significant stretches of alpha-helix don't appear to be involved. So I think it does not follow that the GLXYI motif must itself undergo a conformational change that initiates subsequent events. It is true that its conformation, and likely its conformational plasticity, are probably important. But it's going farther to say that it "induces" conformational responses.

Re: The GYXLI motif (magenta) is located outside the FlhA_C ring and is not directly involved in FlhA_C ring formation. In fact, HS-AFM revealed that FlhA_{C-AAAA} forms a nonameric ring in a way similar to wild-type FlhA_C, indicating that the AAAA mutation does not significantly affect the conformation of the FlhA_C subunit at the inter-subunit interface. Therefore, we suppose that the GGGG mutation does not affect the conformation of the FlhA_C subunit at the inter-subunit interface, either, but inhibits the interaction of FlhA_{L-C} with the D1 and D3 domains of the closest FlhA_C subunit. To avoid over-interpretation, we changed our statement as follow:

Because far-UV CD measurements revealed that the AAAA and GGGG mutations did not severely impair the entire FlhA_C structure (Supplementary Fig. 5b), we suggest that an appropriate conformational change in the GYXLI motif is required for FlhA_{L-C} to bind to the D1 and D3 domains of the closest subunit in the ring.

6. *Sup fig 8; ability to export FlgL without FlgN, in the GGGG or AAAA mutants, is notable. Is this also true for FlgK? Related: The export of FlgK responds to loss of Hl (and the B* mutation) very differently than FlgL. Comments or thoughts on this difference?*

Re: Because no FlgK export was seen in the Δ HI-B* AAAA and Δ HI-B* GGGG mutants (**See Fig 3d, left panel 4th row in the revised manuscript**), we cannot test the effect of FlgN deletion on the export of FlgK in the Δ HI-B* mutant background.

The B* mutation alone does not affect the export properties of the σ^{54} because both FlgK and FlgL are recognized as F-type substrates (**See Supplementary Figure 6b, 4th and 5th rows in the revised manuscript**). But in the absence of FliH and FliI, σ^{54} with the GGGG, AAAA, I372A, or I372G mutation recognizes FlgL as an RH-type substrate while FlgK still being recognized as a F-type substrate. As we responded to Reviewer #2, we have no idea about the reason of this difference.

7. *p. 16 line 19. It is true that B* by itself doesn't change export much. It does not follow that the B* mutation does not play an important role in the changes that are seen when Hl are deleted, though. If involvement of B* is to be ruled out to justify the present focus*

on the role of HI, then the HI deletion needs to be made in the presence of wild-type FlhB. I think the results will be quite different, because the B mutation (I think) is necessary for export to proceed much at all in the absence of HI. If I recall correctly the B* mutation was isolated as a suppressor of the HI deletion, and it seems to make export more permissive. The strengthened phenotypes seen in the HI-B* background are in this sense not surprising, and could be due as much to B* as to loss of HI.*

Re: You are right. The B* mutation has been isolated as a gain-of-function mutation that improves the motility of the *fliH-fliI* double null mutant to a considerable degree (**Minamino and Namba. *Nature* 2008**). The B* mutation increases the probability of the substrate entry into the polypeptide channel of the FT3SS when the extracellular Na⁺ concentration or membrane voltage across the cytoplasmic membrane exceed a certain threshold (**Minamino et al. *Commun. Biol.* 2021; Minamino et al. *PNAS* 2021**). The B* mutation is located near the entrance gate of the polypeptide channel, suggesting that this mutation seems to affect the opening and closing of the entrance gate. Because the B* mutation does not bypass the FliK defect (**unpublished data**), we do not think that the B* mutation is also involved in the export switching function of FlhA with the AAV, AAAT, or GGGV mutation.

8. p. 17 line 17. Can some comment be made about the FlgK row? FlgK export is practically prevented by the delHI-B mutations, and contrasts very strongly with FlgL.*

Re: The FlgN-FlgK chaperone-substrate complex requires FliH and FliI to efficiently bind to FlhA_C for efficient and rapid protein export by the FT3SS (**Minamino et al. *MicrobiologyOpen* 2016**). The AAAA, AAV, GGGG, and GGGV mutations reduced the binding affinity of FlhA_C for the FlgN-FlgK complex (**See Fig. 7 in the revised manuscript**), and so the secretion level of FlgK was significantly reduced in the ΔHI-B* mutant background.

9. p. 17, "with or without B" The statement is true, but pertains to the case with HI present; probably more relevant is whether B* makes any difference when HI is deleted. Likely it does, and in that case, we can't separate the del-HI effect from the B* effect (because B* is always introduced along with delHI in the experiments here).*

Re: Please see our response to your comment #7.

10. p. 20 line 1. Again I wondered why the term 'proofreading' is used.

Re: We deleted the term "proofreading" from the entire text as described above.

11. p. 22, the interpretation of the fliK extragenic suppressors. Alternatively, FlhAc might assist in the FliK-induced dissociation of a part of FlhBc (something that is

thought to occur during the specificity switch!), and the AAAA and GGGG mutations impede this. In this alternative interpretation, the FliK mutations strengthen the ability of FliK to induce the dissociation of the FlhBc bit by itself, for example by slowing the export of FliK so that it has more time to induce the dissociation.*

This raises another: why is the FlhBcc (C-terminal part of the FlhB C-terminal domain) dissociation model not discussed here?

Re: This is a very interesting idea. However, FlhB_{CN} deeply inserts into the hydrophobic core of FlhB_{CC}, and so FlhB_{CN} and FlhB_{CC} tightly associates with each other even after autocleavage as judged by affinity chromatography (**Mianmino and Macnab J. Bacteriol. 2000**). Because we do not have any data suggesting that FlhB_{CC} dissociates from FlhB_{CN} upon hook completion, we cannot discuss the FlhB_{CC} dissociation model at this time.

12. bottom p. 23. The mutation is suggested to restrict a specific process (domain rotation), whereas it seems more likely (to me) that it creates a more permissive, conformationally fluid situation, where a particular, needed conformation is less stabilized than in the wild type.

Re: Agreed and changed “restricts” to “affects”.

REVIEWERS' COMMENTS:

Reviewer #1 (Remarks to the Author):

The revised version of the paper is very satisfactory.
All my concerns have been addressed and the paper is a great contribution to the field.

Reviewer #2 (Remarks to the Author):

This work is a revision of a previous manuscript reviewed by this journal. The authors are investigating a very specific region of the FlhA flagellar type III secretion system export gate protein - small GYXLI motif - for substrate switching. The previous reviews were fairly positive with comments given for clarification and some additional analysis. The authors were very responsive to all the reviewers' comments and the manuscript is improved. I do not have any additional comments or concerns

Reviewer #3 (Remarks to the Author):

The authors have responded constructively to my comments and I feel that the paper is acceptable in its present form.

Our responses are listed below.

To Reviewer #1

The revised version of the paper is very satisfactory. All my concerns have been addressed and the paper is a great contribution to the field.

Re: Thank you so much for all your helpful comments and suggestions.

To Reviewer #2

This work is a revision of a previous manuscript reviewed by this journal. The authors are investigating a very specific region of the FlhA flagellar type III secretion system export gate protein - small GYXLI motif - for substrate switching. The previous reviews were fairly positive with comments given for clarification and some additional analysis. The authors were very responsive to all the reviewers' comments and the manuscript is improved. I do not have any additional comments or concerns.

Re: Thank you so much for all your helpful comments and suggestions.

To Reviewer #3

The authors have responded constructively to my comments and I feel that the paper is acceptable in its present form.

Re: Thank you very much for all your helpful comments and suggestions.